# DEEP METRIC TENSOR REGULARIZED POLICY GRADIENT

## ABSTRACT

In this paper, we propose a novel policy gradient algorithm for deep reinforcement learning. Unlike previous approaches, we focus on leveraging the Hessian trace information in the policy parametric space to enhance the performance of trained policy networks. Specifically, we introduce a metric tensor field that transforms the policy parametric space into a general Riemannian manifold. We further develop mathematical tools, deep learning algorithms, and metric tensor deep neural networks (DNNs) to learn a desirable metric tensor field, with the aim to achieve close-to-zero divergence on the policy gradient vector field of the Riemannian manifold. As an important regularization mechanism, zero divergence nullifies the principal differential components of the loss function used for training policy networks. It is expected to improve the effectiveness and sample efficiency of the policy network training process. Experimental results on multiple benchmark reinforcement learning problems demonstrate the advantages of our metric tensor regularized algorithms over the non-regularized counterparts. Moreover, our empirical analysis reveals that the trained metric tensor DNN can effectively reduce the absolute divergence towards zero on the Riemannian manifold.

## 1 INTRODUCTION

Policy gradient algorithms are an important family of *deep reinforcement learning* (DRL) techniques. They help a DRL agent learn an *optimal policy* that maps any states the agent encounters to optimal actions Schulman et al. (2017); Lillicrap et al. (2015). Unlike Q-learning and other value-based methods, policy gradient algorithms directly train a *deep neural network* (DNN) known as a *policy network* Sutton et al. (2000); Lillicrap et al. (2015). This is achieved by computing the *policy gradient* w.r.t. the trainable parameters of the policy network, known as *policy parameters*, and updating the parameters in the direction of optimizing an agent's *expected cumulative return*.

Many state-of-the-art DRL algorithms rely primarily on the *first-order* information, including policy gradient, to train policy networks Schulman et al. (2017); Fujimoto et al. (2018); Haarnoja et al. (2018). Existing research showed that the estimation of policy gradient has a profound impact on the performance of these algorithms Fujimoto et al. (2018); Wang et al. (2020); Lee et al. (2021). Recently substantial efforts have been made to reduce the bias and variance of the estimated policy gradient Haarnoja et al. (2018); Fan & Ramadge (2021); Zhang et al. (2020). Ensemble learning and hybrid on/off-policy algorithms have also been developed to facilitate reliable estimation of policy gradient for improved exploration and sample efficiency Lee et al. (2021); Januszewski et al. (2021); Chen et al. (2021).

Different from these works, in this paper, we aim to explore the *second-order Hessian* information to train policy networks effectively and efficiently. Several pioneering research works have been reported lately to deepen our understanding of neural networks through the lens of the Hessian, primarily for the supervised learning paradigm Yao et al. (2020); Dong et al. (2020). In the context of DRL, we found that the Hessian information can vary substantially during the training of the policy network. We hypothesize that properly utilizing and controlling the Hessian information can noticeably improve the performance of DRL algorithms.

More concretely, the process of training a policy network can be conceived as an orbit in a high-dimensional *policy parametric space*. Previous research either implicitly or explicitly treated this parametric space as an *Euclidean-like manifold* Martens (2020); Zhang et al. (2019); Kunstner et al.

(2019); Chen et al. (2015, 2014); Peng et al. (2020). Consequently, the *metric tensor field* denoted as $g_{ab}$ on the manifold does not match the differential structure of the policy network and its loss function. Hence, the roughness of the loss function is translated directly to the roughness of the orbit, leading to compromised and unreliable learning performance.

To address this issue, we focus on *Hessian trace* in this paper. In the Euclidean policy parametric space, Hessian trace measures the *divergence* of the vector field w.r.t the policy gradient. Upon generalizing the Euclidean policy parametric space into a *Riemannian manifold*, we propose to achieve close-to-zero divergence as an important regularization mechanism, which helps to nullify the principal differential components of the loss function used for training policy networks Kampffmeyer et al. (2019); Schäfer & Lörch (2019); Liu et al. (2023); Chen (2020). It is hence expected to improve the reliability and effectiveness of the policy network training process.

Driven by this goal, we develop new mathematical tools and DRL algorithms to learn a desirable metric tensor field $g_{ab}$ that induces close-to-zero divergence on the Riemannian manifold. Accordingly, policy network training guided by its *Levi-Civita connection* (aka. *torsion-free $g_{ab}$ compatible derivative operator*) Kreyszig (2013) is expected to be smooth and reliable, resulting in improved effectiveness and sample efficiency.

Notably, $g_{ab}$ is a complex geometric structure, learning which is beyond the capability of existing machine learning models Roy et al. (2018); Le & Cuturi (2015); Beik-Mohammadi et al. (2021). To make $g_{ab}$ regularized DRL feasible and effective, we design a new metric tensor DNN to significantly reduce the complexity involved in learning $g_{ab}$. Specifically, Fourier analysis techniques Rippel et al. (2015) are utilized to reduce the parametric space of the metric tensor DNN. We also propose a parametric matrix representation of high-dimensional special orthogonal groups Gerken et al. (2021); Hutchinson et al. (2021); Chen & Huang (2022) to further simplify the metric tensor DNN by exploiting the symmetries of $g_{ab}$.

The above innovation paves the way to develop a new $g_{ab}$ *regularization algorithm* that uses the learned metric tensor DNN to compute $g_{ab}$ regularized policy gradients for training policy networks. It can be applied to many existing policy gradient algorithms, including Soft Actor Critic (SAC) Haarnoja et al. (2018) and Twin Delayed Deep Deterministic (TD3) Fujimoto et al. (2018). Experiments on multiple benchmark problems confirm that the new $g_{ab}$ regularization algorithm can effectively improve the performance and reliability of SAC and TD3.

**Contributions**: According to our knowledge, we are the first in literature to study mathematical and deep learning techniques to learn $g_{ab}$ and use $g_{ab}$ regularization algorithms to train policy networks. Our research extends the policy parametric space to a general Riemmanian manifold where critical differential geometric information about policy gradients can be captured through the learned $g_{ab}$ and explicitly utilized to boost the learning performance.

## 2 RELATED WORKS

Many recent research works studied a variety of possible ways to estimate policy gradients for effective DRL. For example, Generalized Proximal Policy Optimization (GePPO) introduces a general clipping mechanism to support policy gradient estimation from off-policy samples, achieving a good balance between stability and sample efficiency Queeney et al. (2021). Policy-extended Value Function Approximator (PeVFA) enhances conventional value function approximator by utilizing additional policy representations Tang et al. (2022). This enhancement improves the accuracy of the estimated policy gradients. Efforts have also been made to control the *bias* and *variance* of the estimated policy gradients Fujimoto et al. (2018); Haarnoja et al. (2018); Fan & Ramadge (2021); Zhang et al. (2020). For instance, clipped double Q-learning Fujimoto et al. (2018), entropy regularization Haarnoja et al. (2018), action normalization Wang et al. (2020), and Truncated Quantile Critics (TQC) Kuznetsov et al. (2020) techniques have been proposed to effectively reduce the estimation bias. All of these studies assume that the policy parametric space follows the Euclidean metric and is flat.

The development of natural policy gradient presents a major deviation from the flat policy parametric space Liu et al. (2020); Ding et al. (2020). Its effective use on many challenging DRL problems clearly reveals the importance of expanding the flat policy parametric space to a general Riemannian manifold Grondman et al. (2012). However, since the metric tensor field $g_{ab}$ is defined via the *Fisher information matrix* w.r.t. the policy networks, critical differential geometric information

regarding the DRL problems and their loss functions is not explicitly utilized to boost the learning performance. Using the Fisher information matrix directly to compute the natural policy gradient is also computationally costly in large policy parametric spaces.

In recent literature, notable efforts have been made towards understanding the influence of Hessian information on deep learning models Yao et al. (2020); Dong et al. (2020); Wu et al. (2020); Shen et al. (2019); Singla et al. (2019). For example, efficient numerical linear algebra (NLA) techniques have been developed in Yao et al. (2020) to compute top Hessian eigenvalues, Hessian trace, and Hessian eigenvalue spectral density of DNNs. In Dong et al. (2020), Hessian trace is also exploited to determine suitable quantization scales for different layers of a DNN. Different from these works, instead of examining Hessian information in an Euclidean parametric space, we develop the first time in literature deep learning techniques to alter and improve the differential geometric structure of the policy parametric space.

## 3 BACKGROUND

This paper studies DRL problems modeled as *Markov Decision Processes* (MDPs). An MDP is a tuple $(\mathbb{S}, \mathbb{A}, P, R, \gamma)$ with the *state space* $\mathbb{S}$, the *action space* $\mathbb{A}$, and the *discount factor* $\gamma \in (0, 1]$. The state-transition probability function $P(s, a)$ captures the probability of transiting to any possible next state $s' \sim P(s, a)$ whenever the agent performs action $a \in \mathbb{A}$ in state $s \in \mathbb{S}$. Meanwhile, a scalar reward is determined according to the reward function $R(s, a)$. A *policy* $\pi : \mathbb{S} \to \mathbb{A}$ produces an action $a \in \mathbb{A}$ (or a probability distribution over $\mathbb{A}$) w.r.t. any state $s \in \mathbb{S}$. Its performance is quantified by a value function $v_\pi(s)$ that predicts the *expected cumulative return* obtainable by following $\pi$ to interact with the learning environment, starting from $s \in \mathbb{S}$. The DRL problem has the goal to find an *optimal policy* $\pi^*$ that maximizes its value function w.r.t. any possible initial state $s_0 \in \mathbb{S}$. Such policy is often modeled as a parametric function in the form of a DNN, denoted as $\pi_\theta$, where $\theta \in \mathbb{R}^n$ stands for the $n$-dimensional policy parameter, $n \gg 1$.

## 4 METRIC TENSOR REGULARIZED POLICY GRADIENT

In this section, the $n$-dimensional policy parametric space is transformed into a general Riemannian manifold $(\mathbb{R}^n, g_{ab})$, accompanied by a $(0, 2)$-type metric tensor field $g_{ab}$ defined on $\mathbb{R}^n$ Petersen (2006). We follow the *abstract index notation* commonly used in theoretical physics to represent tensors and their operations Thorne & Blandford (2017). For any policy parameter $\theta \in \mathbb{R}^n$, the tangent vector space at $\theta$ is denoted as $T_\theta$. $g_{ab}$ satisfies two important properties on $T_\theta$, $\forall \theta \in \mathbb{R}^n$:

$$(1) \forall u^a, v^b \in T_\theta, g_{ab}u^a v^b = g_{ba}u^a v^b;$$

$$(2) \forall v^b \in T_\theta, \text{if } u^a \text{ satisfies the equation } g_{ab}u^a v^b = 0, \text{then } u^a = 0.$$

The first property above reveals the *symmetric* nature of $g_{ab}$. The second property requires $g_{ab}$ to be *non-degenerate*. Given any $g_{ab}$ that is $C^\infty$ on $\mathbb{R}^n$, a torsion-free and $g_{ab}$ compatible derivative operator $\nabla_a$ can always be uniquely determined such that $\nabla_a g_{bc} = 0$ on $\mathbb{R}^n$. Unless otherwise specified, $\nabla_a$ always refers to this *compatible derivative operator* in this paper. Using $\nabla_a$, the conventional policy gradient at $\forall \theta \in \mathbb{R}^n$ can be defined as a dual vector of $\theta$ below:

$$\nabla_a \mathbb{E}_{s_0}[v_{\pi_\theta}(s_0)] = \partial_a \mathbb{E}_{s_0}[v_{\pi_\theta}(s_0)] = \sum_{\mu=1}^n \frac{\partial \mathbb{E}_{s_0}[v_{\pi_\theta}(s_0)]}{\partial \theta^{(\mu)}} (\mathrm{d}\theta^{(\mu)})_a,$$

where $\theta^{(\mu)}$ indicates the $\mu$-th dimension of $\theta$. $(\mathrm{d}\theta^{(\mu)})_a$ is the basis dual vector of the dual vector space $T_\theta^*$ at $\theta$. $\partial_a$ is the *ordinary derivative operator*. The policy gradient vector w.r.t. $\nabla_a \mathbb{E}_{s_0}[v_{\pi_\theta}(s_0)]$ is:

$$J^a|_\theta = g^{ab}\nabla_b \mathbb{E}_{s_0}[v_{\pi_\theta}(s_0)] = \sum_{\nu=1}^n \left( \sum_{\mu=1}^n g^{\nu\mu} \frac{\partial \mathbb{E}_{s_0}[v_{\pi_\theta}(s_0)]}{\partial \theta^{(\mu)}} \right) \left( \frac{\partial}{\partial \theta^{(\nu)}} \right)^a,$$

where $(\partial/\partial\theta^{(\nu)})^a$ is the basis vector of the vector space $T_\theta$ at $\theta$. We shall use $J^a|_\theta$ consistently as the *vector representation* of the policy gradient on manifold $(\mathbb{R}^n, g_{ab})$. To obtain $J^a|_\theta$, we need to introduce the *inverse metric tensor* $g^{ab}$ that satisfies

$$g^{ab}g_{bc} = \delta^a_c = \sum_{\mu=1}^n \sum_{\nu=1}^n \delta^\mu_\nu \left( \frac{\partial}{\partial \theta^{(\mu)}} \right)^a (\mathrm{d}\theta^{(\nu)})_c,$$

where $\delta_c^a$ is the $(1,1)$-type *identity tensor* such that $\delta_b^a v^b = v^a, \forall v^a \in T_\theta$, and $\delta_b^a w_a = w_b, \forall w_a \in T_\theta^*$. Accordingly, $\delta_\nu^\mu = 1$ whenever $1 \le \mu = \nu \le n$ and $\delta_\nu^\mu = 0$ otherwise. Given the matrix representation of $g_{ab}$ at any $\theta \in \mathbb{R}^n$ as $G_\theta = [g_{\mu,\nu}(\theta)]_{\mu,\nu=1}^n$, $g^{ab}$ is represented as the inverse matrix $G_\theta^{-1}$. Hence the $g_{ab}$ *regularized policy gradient* can be computed via a matrix expression below

$$\vec{J}|_\theta = G_\theta^{-1} \cdot \nabla_\theta \mathbb{E}_{s_0}[v_{\pi_\theta}(s_0)]. \tag{1}$$

The above vector is called a *vector in linear algebra*. To distinguish it from a vector in differential geometry, we denote it as $\vec{J}$ instead of $J^a$. Each dimension of $\vec{J}$ corresponds to a separate trainable parameter (or dimension) of $\pi_\theta$. The definition of $J^a|_\theta$ (and $\vec{J}|_\theta$) above allows us to construct a vector space of policy gradient on manifold $(\mathbb{R}^n, g_{ab})$, indicated as $J^a$. In differential geometry, *divergence* captures essential information about $J^a$ and is mathematically defined as

$$\forall \theta \in \mathbb{R}^n, Div(J^a)|_\theta = \nabla_a J^a|_\theta.$$

It quantifies the distribution of policy gradient vectors on $(\mathbb{R}^n, g_{ab})$. If the vectors are moving away from $\theta \in \mathbb{R}^n$, the divergence at $\theta$ is positive. If they are converging towards $\theta$, the divergence is negative. When the divergence is close-to-zero, the vectors are neither spreading nor converging at $\theta$. Appendix A shows that achieving close-to-zero divergence can potentially nullify the principal differential components of the loss function used for training $\pi_\theta$.

## 5 METRIC TENSOR REGULARIZATION ALGORITHM FOR TRAINING POLICY NETWORKS

The new $g_{ab}$ regularization algorithm comprises of two components, which will be introduced respectively in Subsections 5.1 and 5.2. We will further apply the $g_{ab}$ regularization method to SAC and TD3 to develop practically useful DRL algorithms in Subsection 5.3.

### 5.1 LEARNING A DNN MODEL OF $g_{ab}$

Let $G_\theta = [g_{\mu,\nu}(\theta)]_{\mu,\nu=1}^n$ be the matrix representation of $g_{ab}$ at any $\theta \in \mathbb{R}^n$. Each entry of this symmetric matrix $G_\theta$, i.e. $g_{\mu,\nu}(\theta)$, is a function of $\theta$. Learning such a matrix representation of $g_{ab}$ directly is a challenging task, since $n \gg 1$ for most of policy networks used by DRL algorithms. To make it feasible to learn $g_{ab}$, we impose a specific structure on $G_\theta$, as given below:

$$G_\theta = I_n + \vec{u}(\theta) \cdot \vec{u}(\theta)^T \tag{2}$$

where $I_n$ stands for the $n \times n$ identity matrix. $\vec{u}(\theta) : \mathbb{R}^n \to \mathbb{R}^n$ is a vector-valued function of $\theta$. Hence $\vec{u}(\theta) \cdot \vec{u}(\theta)^T$ produces an $n \times n$ matrix. It is easy to verify that the simplified matrix $G_\theta$ in equation 2 is symmetric and non-degenerate, suitable to serve as the matrix representation of $g_{ab}$. We aim to learn $g_{ab}$ that can induce zero divergence on the vector field $J^a$ of manifold $(\mathbb{R}^n, g_{ab})$. For this purpose, Proposition 1 below can be utilized to compute the divergence of $J^a$ at any $\theta \in \mathbb{R}^n$.

**Proposition 1** *Given a metric tensor field $g_{ab}$ with its matrix representation defined in equation 2 on manifold $(\mathbb{R}^n, g_{ab})$, the divergence of $C^\infty$ vector field $J^a$ at any $\theta \in \mathbb{R}^n$, i.e. $Div(J^a)|_\theta$, is*

$$Div(J^a)|_\theta = \sum_{\mu=1}^n \left( \frac{\partial \vec{J}^{(\mu)}}{\partial \theta^{(\mu)}} + \frac{\vec{J}^{(\mu)}}{1 + \vec{u}(\theta)^T \cdot \vec{u}(\theta)} \sum_{\nu=1}^n \vec{u}^{(\nu)}(\theta) \frac{\partial \vec{u}^{(\nu)}(\theta)}{\partial \theta^{(\mu)}} \right)$$

*where $\vec{J}^{(\mu)}$ refers to the $\mu$-th dimension of $\vec{J}|_\theta$ at $\theta$. $\theta^{(\mu)}$ and $\vec{u}^{(\nu)}$ represent respectively the $\mu$-th dimension of $\theta$ and $\nu$-th dimension of $\vec{u}(\theta)$. A proof of Proposition 1 is given in Appendix B.*

While $\vec{u}(\theta)$ in equation 2 can be arbitrary functions of $\theta$, to tackle the complexity of learning $\vec{u}(\theta)$, we can re-formulate $\vec{u}(\theta)$ in the form of a parameterized linear transformation of $\theta$, i.e.

$$\vec{u}(\theta) = S(\theta, \phi_1) R(\theta, \phi_2) \theta \tag{3}$$

where $S(\theta, \phi_1)$ stands for the $n \times n$ *scaling* (diagonal) matrix w.r.t. $\theta$ and parameterized by $\phi_1$. $R(\theta, \phi_2)$ stands for the $n \times n$ *rotation matrix* w.r.t. $\theta$ and parameterized by $\phi_2$. Meanwhile,

$dim(\phi_1) + dim(\phi_2) = m \ll n$. $S(\theta, \phi_1)$ and $R(\theta, \phi_2)$ together define a linear transformation of $\theta$ that involves the two fundamental operations, i.e. *scaling* and *rotation*.

Concretely, $S(\theta, \phi_1) = Diag(\vec{\omega}(\theta, \phi_1))$ controls the magnitude of each dimension of $\vec{u}(\theta)$. The diagonal line of matrix $S(\theta, \phi_1)$ forms an $n$-dimensional vector $\vec{\omega}(\theta, \phi_1)$. While it sounds straightforward to let $\vec{\omega}(\theta, \phi_1) = \phi_1$, this implies that $dim(\phi_1) = n$, contradicting with the requirement that $m \ll n$. To tackle this issue, we perform Fourier transformation of $\vec{\omega}$ and only keep the low-frequency components of $\vec{\omega}$ which can be further controlled via $\phi_1$. Specifically, define a series of $n$-dimensional vectors $\vec{\Omega}^{(i)}$ using the trigonometrical function $cos()$ as

$$\vec{\Omega}^{(i)} = \sqrt{\frac{2}{n}} \left[ \begin{array}{c} cos\left(\frac{2\pi i}{n} j\right) |_{j=0} \\ \vdots \\ cos\left(\frac{2\pi i}{n} j\right) |_{j=n-1} \end{array} \right],$$

where $1 \leq i \leq \tilde{m}$. Further define $\Omega$ as an $n \times \tilde{m}$ matrix:

$$\Omega = [\vec{\Omega}^{(1)}, \ldots, \vec{\Omega}^{(\tilde{m})}]$$

Then $\vec{\omega}(\theta, \phi_1)$ can be obtained through the matrix expression below:

$$\vec{\omega}(\theta, \phi_1) = \Omega \cdot \vec{\tilde{w}}(\theta, \phi_1), \tag{4}$$

where the parameterized $\tilde{m}$-dimensional vector $\vec{\tilde{w}}(\theta, \phi_1)$ controls the magnitude of the $\tilde{m}$ low-frequency components of $\vec{\omega}$. Consequently, the problem of learning the $n \times n$ scaling matrix $S(\theta, \phi_1)$ is reduced to the problem of learning $\phi_1$ at $\theta \in \mathbb{R}^n$ with $dim(\phi_1) \ll n$.

In group theory, any $n \times n$ rotation matrix serves as the matrix representation of a specific element of the $n$-dimensional *Special Orthogonal* (SO) group, denoted as $SO(n)$ Hall (2013). Consider the Lie algebra of $SO(n)$, indicated as $\mathcal{SO}(n)$. $\mathcal{SO}(n)$ is defined mathematically below

$$\mathcal{SO}(n) = \{n \times n \text{ real-valued matrix } A | A^T = -A\}.$$

In other words, $\mathcal{SO}(n)$ is the set of all $n \times n$ *anti-symmetric matrices*. Consequently, $\forall A \in \mathcal{SO}(n)$, $\exp(A)$ must be an $n \times n$ rotation matrix. In view of this, we further introduce Proposition 2 below to simplify the parameterization of $R(\theta, \phi_2)$. Its proof is given in Appendix C.

**Proposition 2** $\forall A \in \mathcal{SO}(n)$, *there exist* $n \times n$ *unitary matrices* $U$ *and* $V$ *such that*

$$\exp(A) = U \cdot \Sigma_c \cdot U^T - V \cdot \Sigma_s \cdot U^T$$

*where, w.r.t. an $n$-dimensional vector $\vec{\sigma} = [\sigma^{(1)}, \ldots, \sigma^{(n)}]^T$, $\Sigma_c$ and $\Sigma_s$ are defined respectively as*

$$\Sigma_c = \left[ \begin{array}{ccc} cos(\sigma^{(1)}) & & 0 \\ & \ddots & \\ 0 & & cos(\sigma^{(n)}) \end{array} \right] \text{ and } \Sigma_s = \left[ \begin{array}{ccc} sin(\sigma^{(1)}) & & 0 \\ & \ddots & \\ 0 & & sin(\sigma^{(n)}) \end{array} \right]$$

Following Proposition 2, we can construct $R(\theta, \phi_2)$. Notice that

$$(\vec{\Omega}^{(i)})^T \cdot \vec{\Omega}^{(j)} \approx \left\{ \begin{array}{ll} 1, & i = j \\ 0, & i \neq j \end{array} \right. , \forall i, j \in \{1, \ldots, \tilde{m}\}$$

$\Omega$ can be utilized to approximate the first unitary matrix $U$ in Proposition 2. Similarly, we can define another series of $n$-dimensional vectors $\vec{\Phi}^{(i)}$ as

$$\vec{\Phi}^{(i)} = \sqrt{\frac{2}{n}} \left[ \begin{array}{c} sin\left(\frac{2\pi i}{n} j\right) |_{j=0} \\ \vdots \\ sin\left(\frac{2\pi i}{n} j\right) |_{j=n-1} \end{array} \right],$$

where $1 \leq i \leq \tilde{m}$. $\Phi = [\vec{\Phi}^{(1)}, \ldots, \vec{\Phi}^{(\tilde{m})}]$ gives a good approximation of the second unitary matrix $V$ in Proposition 2. However, different from $U$ and $V$, which are $n \times n$ matrices, $\Omega$ and $\Phi$ are $n \times \tilde{m}$

matrices. To cope with this difference in dimensionality, we introduce a parameterized $\tilde{m}$-dimensional vector $\vec{\tilde{\sigma}}(\theta, \phi_2)$. Assume that functions $cos()$ and $sin()$ are applied elementary-wise to $\vec{\tilde{\sigma}}(\theta, \phi_2)$, then

$$\tilde{\Sigma}_c = Diag(cos(\vec{\tilde{\sigma}}(\theta, \phi_2))) \text{ and } \tilde{\Sigma}_s = Diag(sin(\vec{\tilde{\sigma}}(\theta, \phi_2)))$$

are $\tilde{m} \times \tilde{m}$ diagonal matrices. Subsequently, define $n \times n$ matrix

$$\tilde{R}(\theta, \phi_2) = \Omega \cdot \tilde{\Sigma}_c \cdot \Omega^T - \Phi \tilde{\Sigma}_s \cdot \Omega^T. \tag{5}$$

Similar to the construction of the scaling matrix, equation 5 also draws inspiration from frequency analysis, as clearly revealed by Proposition 3 below, which is proved in Appendix D.

**Proposition 3** *Given $A \in \mathcal{SO}(n)$, assume that $\exp(A) = \hat{\Omega} \cdot \Sigma_c \cdot \hat{\Omega}^T - \hat{\Phi} \cdot \Sigma_s \cdot \hat{\Omega}^T$ as in Proposition 2, where $\hat{\Omega}$ and $\hat{\Phi}$ are defined similarly as $\Omega$ and $\Phi$ with the additional requirement that $\tilde{m} = n$. Hence $\hat{\Omega}$ and $\hat{\Phi}$ are $n \times n$ unitary matrices. Under this assumption, for any n-dimensional vector $\vec{a}$,*

$$\exp(A) \cdot \vec{a} = \sum_{i=1}^{n} \eta_i \sqrt{\frac{2}{n}} \left[ \begin{array}{c} cos\left(\frac{2\pi i}{n} j + \vec{\sigma}^{(i)}|_{j=0}\right) \\ \vdots \\ cos\left(\frac{2\pi i}{n} j + \vec{\sigma}^{(i)}|_{j=n-1}\right) \end{array} \right]$$

*where $\eta_i = (\vec{\hat{\Omega}}^{(i)})^T \cdot \vec{a}$ stands for the magnitude of the i-th frequency component of $\vec{a}$[1]*

Proposition 3 indicates that, upon multiplying the rotation matrix $\exp(A)$ with any vector $\vec{a}$, this will result in independent phase shift of each frequency component of $\vec{a}$, controlled by the respective dimension of vector $\vec{\sigma}$ in Proposition 2. Therefore, $\tilde{R}(\theta, \phi_2)$ in equation 5 only shifts/rotates the first $\tilde{m}$ low frequency components of a vector upon multiplying it with the vector. In view of this, a full-ranked parameterized rotation matrix can be constructed as

$$R(\theta, \phi_2) = \tilde{R}(\theta, \phi_2) + I_n - \Omega \cdot \Omega^T. \tag{6}$$

Whenever $R(\theta, \phi_2)$ in equation 6 is multiplied with any vector $\vec{a}$, only the low-frequency components of $\vec{a}$ is phase shifted/rotated. The high-frequency components of $\vec{a}$ remain untouched. Subsequently, the problem of learning the $n \times n$ rotation matrix $R(\theta, \phi_2)$ is reduced to the problem of learning the $\tilde{m}$-dimensional vector $\vec{\tilde{\sigma}}(\theta, \phi_2)$ parameterized by $\phi_2$.

Given the parameterized model of $G_\theta$ based on equation 3, equation 4, equation 5 and equation 6 and using Proposition 1, the problem of learning $g_{ab}$ can be formulated as an optimization problem:

$$\min_{\phi_1,\phi_2}(Div(J^a)|_\theta)^2 = \min_{\phi_1,\phi_2}\left(\sum_{\mu=1}^{n}\left(\frac{\partial \vec{J}^{(\mu)}}{\partial \theta^{(\mu)}} + \frac{\vec{J}^{(\mu)}}{1 + \vec{u}(\theta,\phi)^T \cdot \vec{u}(\theta,\phi)}\sum_{\nu=1}^{n}\vec{u}^{(\nu)}(\theta,\phi)\frac{\partial \vec{u}^{(\nu)}(\theta,\phi)}{\partial \theta^{(\mu)}}\right)\right)^2 \tag{7}$$

Driven by this problem, $\phi_1$ and $\phi_2$ can be repeatedly updated towards minimizing $(Div(J^a)|_\theta)^2$, so as to bring the divergence of $J^a$ close to 0. For this purpose, we design a *metric tensor DNN* (see Appendix G) that processes $\theta$ as its input and produces $\vec{\tilde{\omega}}(\theta, \phi_1)$ and $\vec{\tilde{\sigma}}(\theta, \phi_2)$ as its output. $\phi_1$ and $\phi_2$ are the trainable parameters of this DNN.

## 5.2 USING LEARNED $g_{ab}$ MODEL TO COMPUTE REGULARIZED POLICY GRADIENT

Using the metric tensor DNN as a deep model of $g_{ab}$, we develop two alternative methods to compute $g_{ab}$ regularized policy gradient. The **first method** directly follows equation 1. Specifically, according to the Sherman-Morrison formula Press et al. (2007),

$$G_\theta^{-1} = I_n - \frac{\vec{u}(\theta,\phi) \cdot \vec{u}(\theta,\phi)^T}{1 + \vec{u}(\theta,\phi)^T \cdot \vec{u}(\theta,\phi))}$$

Consequently,

$$\vec{J}|_\theta = \nabla_\theta \mathbb{E}_{s_0}[v_{\pi_\theta}(s_0)] - \frac{\vec{u}(\theta,\phi)^T \cdot \nabla_\theta \mathbb{E}_{s_0}[v_{\pi_\theta}(s_0)]}{1 + \vec{u}(\theta,\phi)^T \cdot \vec{u}(\theta,\phi))}\vec{u}(\theta,\phi) \tag{8}$$

---

[1]Vector $\vec{a}$ in Proposition 3 is treated as a signal indexed by its dimensions.

The **second method** aims to update $\theta$ along the direction of the *geodesic* Kreyszig (2013), which is jointly and uniquely determined by the learned $g_{ab}|_\theta$ and $\vec{J}|_\theta$. Geodesics generalize straight lines for solving optimization problems on high-dimensional manifolds $(\mathbb{R}^n, g_{ab})$ Hu et al. (2020). For simplicity and clarity, we use the term *geodesic regularized policy gradient* $\vec{T}|_\theta$ to indicate the direction of the geodesic at $\theta$, in order to clearly distinguish it from $g_{ab}$ regularized policy gradient $\vec{J}|_\theta$ in equation 8. Proposition 4 provides an efficient way to estimate $\vec{T}|_\theta$.

**Proposition 4** *Given the manifold $(\mathbb{R}^n, g_{ab})$ of the policy parametric space, at any $\theta \in \mathbb{R}^n$, a geodesic $\Gamma$ that passes through $\theta$ can be uniquely and jointly determined by $g_{ab}$ and the $g_{ab}$ regularized policy gradient vector $J^a|_\theta$ at $\theta$. Assume that $g_{ab}$ changes smoothly and stably along $\Gamma$[2], there exist $\zeta_1, \zeta_2 > 0$ such that the geodesic regularized policy gradient at $\theta$ can be approximated as*

$$\vec{T}^{(\delta)}|_\theta \approx (1 + \zeta_1)\vec{J}^{(\delta)}|_\theta + \zeta_2 \sum_{\rho=1}^{n} g^{\delta\rho}(\theta) \sum_{\mu=1}^{n} \sum_{\nu=1}^{n} \frac{\partial g_{\mu\nu}(\theta)}{\partial \theta^{(\rho)}} \vec{J}^{(\mu)}|_\theta \vec{J}^{(\nu)}|_\theta$$

*where $\vec{T}^{(\delta)}|_\theta$ stands for the $\delta$-th dimension of the geodesic regularized policy gradient $\vec{T}$ at $\theta$, $0 \le \delta \le n$. A proof of this proposition is given in Appendix E.*

$\vec{T}|_\theta$ in Proposition 4 is obtained by updating $\vec{J}|_\theta$ with a new term controlled by $\zeta_2$. We treat $\frac{\zeta_2}{1+\zeta_1}$ as a hyper-parameter of our $g_{ab}$ regularization algorithm to adjust the influence of this new term.

### 5.3 DRL ALGORITHMS BASED ON $g_{ab}$ REGULARIZED POLICY GRADIENT

Following the mathematical and algorithmic developments in Subsections 5.1 and 5.2, a new $g_{ab}$ regularization algorithm is designed to compute $g_{ab}$ regularized policy gradients, as presented in Algorithm 1 and further explained in Appendix F. Building on Algorithm 1, we can modify existing DRL algorithms to construct their $g_{ab}$ regularized counterparts. We specifically considered two DRL algorithms, namely SAC and TD3, due to their widespread popularity Haarnoja et al. (2018); Fujimoto et al. (2018). It remains as an important future work to study the effective use of Algorithm 1 in other DRL algorithms. Algorithm 2 in Appendix F presents the details of $g_{ab}$ regularized DRL algorithms. Following Algorithm 2, we can identify four algorithm variants, including SAC-J and TD3-J that use $g_{ab}$ regularized policy gradients, as well as SAC-T and TD3-T that use geodesic regularized policy gradients. All these variants are experimentally examined in Section 6.

## 6 EXPERIMENTS

**Implementation:** We use the popular OpenAI Spinning Up repository Achiam (2018) to implement $g_{ab}$ regularized DRL algorithms introduced in the previous section. To learn the complex geometric structure of $g_{ab}$, we introduce a new metric tensor DNN architecture parameterized by both $\phi_1$ and $\phi_2$ in Appendix G. It transforms the $n$-dimensional policy parameter of a policy network $\pi_\theta$ into two $\tilde{m}$-dimensional vectors $\vec{\tilde{\omega}}(\theta, \phi_1)$ and $\vec{\tilde{\sigma}}(\theta, \phi_2)$, which are used to build the scaling matrix $S(\theta, \phi_1)$ and the rotation matrix $R(\theta, \phi_2)$ in equation 3 respectively.

Our implementation follows closely all hyper-parameter setting and network architectures reported in Haarnoja et al. (2018); Fujimoto et al. (2018). Since calculating the Hessian trace precisely can pose significant computation burden on existing deep learning libraries such as PyTorch, we adopt a popular Python library named PyHessian Yao et al. (2020), where Hutchinson's method Avron & Toledo (2011); Bai et al. (1996) is employed to estimate the Hessian trace efficiently. See Appendix H for the detailed experiment setup.

Experiments have been conducted on multiple challenging continuous control benchmark problems provided by OpenAI Gym Brockman et al. (2016) and PyBullet Ellenberger (2018–2019). Each benchmark problem has a maximum episode length of 1000 timesteps. Each DRL algorithm is trained for $300k$ timesteps. To obtain the cumulative returns, we average the results of 10 independent testing episodes after every 1000 training timesteps for each individual algorithm run. Every competing algorithm was also run for 5 different seeds to determine its average performance.

---

[2]See Appendix E for the precise definition of this assumption.

**Performance Comparison:** The comparison between SAC and its metric tensor regularized variations, SAC-J and SAC-T, is presented in Table 1 and Figure 1. As indicated in the table, SAC-T outperforms both SAC and SAC-J on majority of the benchmark problems, except Walker2D-v0, where SAC-T achieved 99% of the highest cumulative returns obtained by SAC. Furthermore, in the case of the Walker2D-v3 problem, SAC-T achieved on average over 60% higher cumulative returns in comparison to SAC and over 100% higher cumulative returns when compared to SAC-J.

Similar results can be observed upon comparing TD3, TD3-J, and TD3-T. As reported in Table 1 and Figure 1, although TD3-T only achieved over 85% of the highest cumulative returns obtained by TD3-J on AntPyBullet and Walker2D-v3, TD3-T doubled the cumulative returns compared to TD3 and TD3-J on the InvertedDoublePendulum-v2 problem. This observation not only supports our previous findings but also demonstrates the broad applicability of our proposed metric tensor regularization algorithm.

We also found that using $g_{ab}$ regularized policy gradient alone may not always lead to noticeable performance gains since SAC-J outperformed SAC on two benchmark problems (i.e. Ant-v0 and InvertedDoublePendulum-v2) but also performed worse on two benchmark problems (i.e. Walker2D-v3 and Walker2D-v0). These results suggest that it is more desirable to train policy parameters in the direction of the geodesics in a general Rimannian manifold $(\mathbb{R}^n, g_{ab})$ in order for $g_{ab}$ regularized policy gradient to effectively improve the performance of DRL algorithms. This observation agrees well with existing optimization techniques on Rimennian manifolds Hu et al. (2020).

Table 1: Final performance of competing algorithms on 4 benchmark problems after 300k timesteps.

| Benchmark problems | SAC | SAC-J | SAC-T | TD3 | TD3-J | TD3-T |
|---|---|---|---|---|---|---|
| InvertedDoublePendulum-v2 (Mujoco) | 9312.77 | 9356.47 | **9356.91** | 3129.28 | 4679.69 | **8731.77** |
| Walker2D-v3 (Mujoco) | 1689.15 | 1290.35 | **2762.51** | 3325.71 | **3879.81** | 3333.42 |
| Ant-v0 (PyBullet) | 780.69 | 798.89 | **837.47** | 2734.56 | **2848.43** | 2754.12 |
| Walker2D-v0 (PyBullet) | **945.72** | 905.28 | 938.68 | 1327.33 | 1364.34 | **1727.88** |

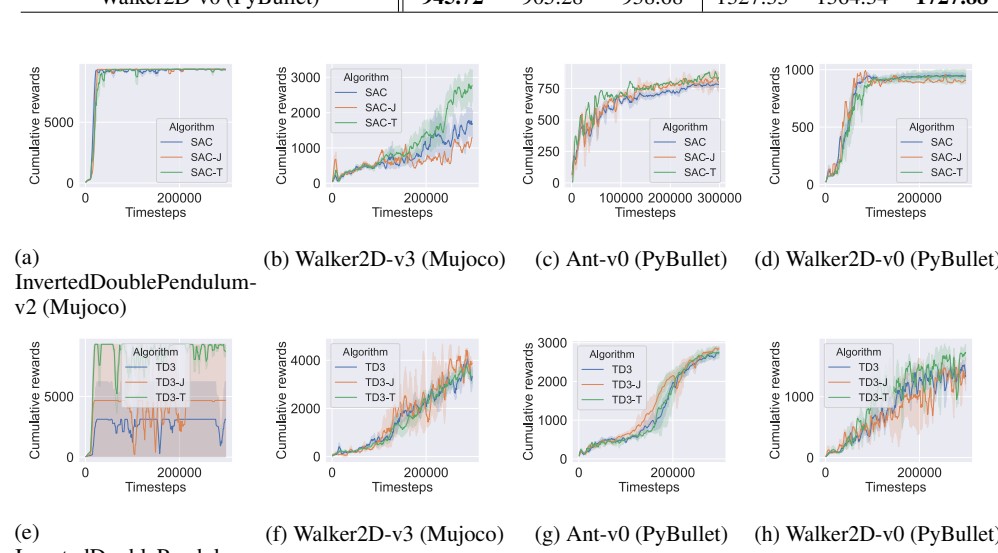

(a) InvertedDoublePendulum-v2 (Mujoco)

(b) Walker2D-v3 (Mujoco)

(c) Ant-v0 (PyBullet)

(d) Walker2D-v0 (PyBullet)

(e) InvertedDoublePendulum-v2 (Mujoco)

(f) Walker2D-v3 (Mujoco)

(g) Ant-v0 (PyBullet)

(h) Walker2D-v0 (PyBullet)

Figure 1: Learning curves of SAC, TD3 and their metric tensor regularized variants on four benchmark RL problems over 300k timsteps.

**Further analysis of the metric tensor learning technique:** We experimentally show the effectiveness of using the proposed metric tensor DNN to learn $g_{ab}|_{\theta}$ at any $\theta \in \mathbb{R}^n$ so that $|Div(J^a)|_{\theta}|$ can be made closer to zero. For this purpose, we introduce a new quantity named the *divergence ratio*. It is defined as the *absolute ratio* between the divergence of $J^a$ in the general manifold $(\mathbb{R}^n, g_{ab})$ and the Hessian trace of the policy gradient, which is the divergence of $J^a$ in the Euclidean policy parametric space $(\mathbb{R}^n, \delta_{ab})$. $\delta_{ab}$ is the identity metric tensor.

The divergence ratio quantifies the relative divergence changes upon extending the Euclidean policy parametric space into a general Riemannian manifold with the introduction of the metric tensor field $g_{ab}$. Specifically, whenever the divergence ratio is less than 1 and close to 0, the absolute divergence $|Div(J^a)|_\theta|$ in the manifold $(\mathbb{R}^n, g_{ab})$ is smaller than the absolute divergence in the Euclidean policy parametric space, implying that the policy gradient vector field becomes smoother in the manifold $(\mathbb{R}^n, g_{ab})$. As demonstrated by the above experiment results, this is expected to allow policy network training to be performed effectively and stably. On the other hand, if the divergence ratio is above 1, it indicates that the policy gradient vector field becomes less smooth in the manifold $(\mathbb{R}^n, g_{ab})$. In this case, our metric tensor regularized policy gradient algorithms will resort to using normal policy gradients in the Euclidean policy parametric space to train policy networks.

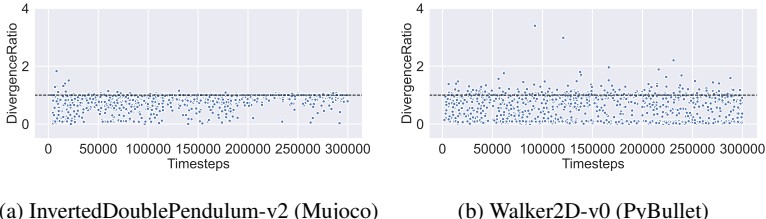

(a) InvertedDoublePendulum-v2 (Mujoco)      (b) Walker2D-v0 (PyBullet)

Figure 2: Divergence ratios obtained by TD3-T during the training process, where the divergence ratio is defined as the absolute ratio between $Div(J^a)|_\theta$ and the Hessian trace.

Figure 2 presents the divergence ratios obtained by TD3-T on two benchmark problems. Evidenced by the figure, using the trained metric tensor DNN and the corresponding $g_{ab}$, TD3-T successfully reduces a significant portion of the divergence ratios to below 1 during the training process. Over 75% of the divergence ratios obtained by TD3-T during policy training are less than 1 on both benchmark problems. Detailed experiment results can be found in Appendix I. Our results demonstrate the effectiveness of using the metric tensor regularization algorithm to train the metric tensor DNN to achieve close-to-zero divergence on the policy parametric space.

We further present the Hessian trace obtained by SAC and TD3 on several benchmark problems respectively in Appendix J. Interestingly, the results show that the Hessian trace obtained by using the same algorithm such as SAC-T can vary greatly on different benchmark problems. Meanwhile, even on the same benchmark problem, the Hessian traces produced by different algorithms such as SAC-T and TD3-T can be significantly different. Driven by this observation, we believe the impact of Hessian trace on the performance of policy gradient algorithms should never be neglected. Our metric tensor regularized policy gradients present the first attempt in the literature towards utilizing and controlling the Hessian trace for effective training of policy networks. Finally, sensitivity analysis of three key hyper-parameters of our new algorithms is reported in Appendix K.

## 7 CONCLUSIONS

In this paper, we studied policy gradient techniques for deep reinforcement learning. Most of the existing policy gradient algorithms rely primarily on the first-order policy gradient information to train policy networks. We developed new mathematical and deep learning techniques to effectively utilize and control the Hessian trace associated with the policy gradient, in order to improve the performance of these algorithms. Hessian trace gives the divergence of the policy gradient vector field on the Euclidean policy parametric space. We can effectively reduce the absolute divergence towards zero so as to smoothen the policy gradient vector field. This was achieved by using our newly developed mathematical and deep learning techniques and our metric DNN in this paper. Armed with these new technical developments, we have further created new metric tensor regularized policy gradient algorithms based on SAC and TD3. The newly proposed algorithms were evaluated experimentally on several benchmark RL problems. Our experiment results confirmed that the new algorithms can significantly outperform their counterparts that do not use our metric tensor regulization techniques. Additional experiment results also confirmed that the trained metric tensor DNN in our algorithms can effectively reduce the absolute divergence towards zero on the general Riemmanian manifold.

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
