## Appendix A

In this appendix, we demonstrate the potential advantages of achieving zero (or close-to-zero) divergence everywhere on $(\mathbb{R}^n, g_{ab})$.

For any $\theta \in \mathbb{R}^n$, we can perform a second-order Taylor expansion of $\mathbb{E}_{s_0}[v_{\pi_\theta}(s_0)]$ on the manifold $(\mathbb{R}^n, g_{ab})$, as presented below:

$$\mathbb{E}_{s_0}[v_{\pi_{(\theta+\vec{v})}}(s_0)] \approx \mathbb{E}_{s_0}[v_{\pi_\theta}(s_0)] + v^a \nabla_a \mathbb{E}_{s_0}[v_{\pi_\theta}(s_0)] + \frac{1}{2} v^a v^b \nabla_a \nabla_b \mathbb{E}_{s_0}[v_{\pi_\theta}(s_0)] \qquad (9)$$

Here $v^a$ refers to an arbitrary vector at $\theta$ that can cause a small positional change of $\theta$ in the policy parametric space. We use $\vec{v} \in \mathbb{R}^n$ to indicate the same vector in linear algebra. Hence $\theta + \vec{v}$ corresponds to an different element of the manifold $(\mathbb{R}^n, g_{ab})$ that is close to $\theta$. Note that $v^a = g^{ab} v_b$, hence

$$\frac{1}{2} v^a v^b \nabla_a \nabla_b \mathbb{E}_{s_0}[v_{\pi_\theta}(s_0)] = \frac{1}{2} v_b v^b g^{ab} \nabla_a \nabla_b \mathbb{E}_{s_0}[v_{\pi_\theta}(s_0)]$$

If the divergence of $J^a$ is 0 (or close-to-zero) at $\theta$, by definition $\nabla_b (g^{ab} \nabla_a \mathbb{E}_{s_0}[v_{\pi_\theta}(s_0)])|_\theta = 0$. Therefore

$$\nabla_b (g^{ab} \nabla_a \mathbb{E}_{s_0}[v_{\pi_\theta}(s_0)])|_\theta = g^{ab} \nabla_a \nabla_b \mathbb{E}_{s_0}[v_{\pi_\theta}(s_0)])|_\theta = 0.$$

In other words, by jointly considering all $n$ dimensions of $\theta$, we have

$$\sum_{\mu=1}^{n} \sum_{\nu=1}^{n} g^{\mu\nu} \nabla_{\theta^{(\mu)}} \nabla_{\theta^{(\nu)}} \mathbb{E}_{s_0}[v_{\pi_\theta}(s_0)]) = 0.$$

Assume that $v^a$ satisfies $v^{(\mu)} v_{(\mu)} = c^2, \forall \mu = 1, \ldots, n$, where $c$ is a real-valued constant. $v^{(\mu)}$ and $v_{(\mu)}$ refer respectively to the $\mu$-th dimension of vector $v^a$ and its corresponding dual vector $v_a$ at $\theta$. In other words, the magnitude of change caused by $v^a$ on $\theta$ stays at a fixed level across all dimensions. Using this condition, we can simplify the Taylor expansion in equation 9 as:

$$\mathbb{E}_{s_0}[v_{\pi_{(\theta+\vec{v})}}(s_0)]$$
$$\approx \mathbb{E}_{s_0}[v_{\pi_\theta}(s_0)] + \sum_{\mu=1}^{n} v^{(\mu)} \frac{\partial \mathbb{E}_{s_0}[v_{\pi_\theta}(s_0)]}{\partial \theta^{(\mu)}} + \frac{1}{2} \sum_{\mu=1}^{n} \sum_{\nu=1}^{n} v^\nu v_\nu g^{\mu\nu} \nabla_{\theta^{(\mu)}} \nabla_{\theta^{(\nu)}} \mathbb{E}_{s_0}[v_{\pi_\theta}(s_0)])$$
$$= \mathbb{E}_{s_0}[v_{\pi_\theta}(s_0)] + \sum_{\mu=1}^{n} v^{(\mu)} \frac{\partial \mathbb{E}_{s_0}[v_{\pi_\theta}(s_0)]}{\partial \theta^{(\mu)}} + \frac{c^2}{2} \sum_{\mu=1}^{n} \sum_{\nu=1}^{n} g^{\mu\nu} \nabla_{\theta^{(\mu)}} \nabla_{\theta^{(\nu)}} \mathbb{E}_{s_0}[v_{\pi_\theta}(s_0)])$$
$$= \mathbb{E}_{s_0}[v_{\pi_\theta}(s_0)] + \sum_{\mu=1}^{n} v^{(\mu)} \frac{\partial \mathbb{E}_{s_0}[v_{\pi_\theta}(s_0)]}{\partial \theta^{(\mu)}}.$$

Therefore, with zero (or close-to-zero) divergence, the second-order differential components involved in approximating $\mathbb{E}_{s_0}[v_{\pi_{(\theta+\vec{v})}}(s_0)]$ can be nullified. Since the above approximation guides the training of policy networks in practice, $g_{ab}$ regularized policy gradient in equation 1 has the potential to improve the performance and sample efficiency of policy gradient based DRL algorithms. Driven by this motivation, we aim to develop mathematical tools and deep learning techniques to achieve $g_{ab}$ regularized policy network training in this paper.

## Appendix B

This appendix presents a proof of Proposition 1. The divergence of $C^\infty$ vector field $J^a|_\theta$ at any $\theta \in \mathbb{R}^n$ satisfies the equation below:

$$Div(J^a)|_\theta = \nabla_a J^a = \frac{1}{\sqrt{|g|}} \sum_{\mu=1}^{n} \frac{\partial}{\partial \theta^{(\mu)}} \left( \sqrt{|g|} \vec{J}^{(\mu)} \right)$$

where $g = det(G_\theta)$. Following the specific structure of $G_\theta$ in equation 2 and using the matrix determinant lemma Press et al. (2007),

$$g = 1 + \vec{u}^T(\theta, \phi) \cdot \vec{u}(\theta, \phi) > 0$$

Hence, $\sqrt{|g|} = \sqrt{g}$. Let

$$\epsilon = \sum_{\mu=1}^{n} \frac{\partial}{\partial \theta^{(\mu)}} \left( \sqrt{|g|} \vec{J}^{(\mu)} \right)$$

$$= \sqrt{g} \sum_{\mu=1}^{n} \left( \frac{\partial \vec{J}^{(\mu)}}{\partial \theta^{(\mu)}} + \frac{\vec{J}^{(\mu)}}{\sqrt{g}} \frac{\partial \sqrt{g}}{\partial \theta^{(\mu)}} \right)$$

Using Jacobi's formula Bellman (1997) below

$$\frac{\partial}{\partial \theta^{(\mu)}} det(G_\theta) = det(G_\theta) Tr \left( G_\theta^{-1} \frac{\partial G_{(\theta)}}{\partial \theta^{(\mu)}} \right),$$

$\epsilon$ can be re-written as

$$\epsilon = \sqrt{g} \sum_{\mu=1}^{n} \left( \frac{\partial \vec{J}^{(\mu)}}{\partial \theta^{(\mu)}} + \frac{\vec{J}^{(\mu)}}{2} Tr \left( G_\theta^{-1} \frac{\partial G_{(\theta)}}{\partial \theta^{(\mu)}} \right) \right)$$

Notice that

$$\frac{\partial G_\theta}{\partial \theta^{(\mu)}} = \left( \frac{\partial \vec{u}(\theta, \phi)}{\partial \theta^{(\mu)}} \right) \cdot \vec{u}(\theta, \phi)^T + \vec{u}(\theta, \phi) \cdot \left( \frac{\partial \vec{u}(\theta, \phi)}{\partial \theta^{(\mu)}} \right)^T$$

Clearly there are two parts in the above equation. We refer to them respectively as $P1(\frac{\partial G_\theta}{\partial \theta^{(\mu)}})$ and $P2(\frac{\partial G_\theta}{\partial \theta^{(\mu)}})$. Using these notations,

$$G_\theta^{-1} P2 \left( \frac{\partial G_\theta}{\partial \theta^{(\mu)}} \right) = \vec{u}(\theta, \phi) \cdot \left( \frac{\partial \vec{u}(\theta, \phi)}{\partial \theta^{(\mu)}} \right)^T - \frac{\vec{u}(\theta, \phi) \cdot \vec{u}(\theta, \phi)^T}{1 + \vec{u}(\theta, \phi)^T \cdot \vec{u}(\theta, \phi)} \vec{u}(\theta, \phi) \left( \frac{\partial \vec{u}(\theta, \phi)}{\partial \theta^{(\mu)}} \right)^T$$

$$= \frac{1}{\vec{u}(\theta, \phi)^T \cdot \vec{u}(\theta, \phi)} \vec{u}(\theta, \phi) \left( \frac{\partial \vec{u}(\theta, \phi)}{\partial \theta^{(\mu)}} \right)^T$$

Meanwhile,

$$G_\theta^{-1} P1 \left( \frac{\partial G_\theta}{\partial \theta^{(\mu)}} \right) = \left( \frac{\partial \vec{u}(\theta, \phi)}{\partial \theta^{(\mu)}} \right) \cdot \vec{u}(\theta, \phi)^T - \frac{\vec{u}(\theta, \phi) \cdot \vec{u}(\theta, \phi)^T}{1 + \vec{u}(\theta, \phi)^T \cdot \vec{u}(\theta, \phi)} \left( \frac{\partial \vec{u}(\theta, \phi)}{\partial \theta^{(\mu)}} \right) \cdot \vec{u}(\theta, \phi)^T$$

$$= \left( \frac{\partial \vec{u}(\theta, \phi)}{\partial \theta^{(\mu)}} \right) \cdot \vec{u}(\theta, \phi)^T - \frac{\vec{u}(\theta, \phi)^T \cdot \frac{\partial \vec{u}(\theta, \phi)}{\partial \theta^{(\mu)}}}{1 + \vec{u}(\theta, \phi)^T \cdot \vec{u}(\theta, \phi)} \vec{u}(\theta, \phi) \cdot \vec{u}(\theta, \phi)^T$$

Subsequently,

$$Tr \left( G_\theta^{-1} \frac{\partial G_\theta}{\partial \theta^{(\mu)}} \right) = Tr \left( \frac{1}{\vec{u}(\theta, \phi)^T \cdot \vec{u}(\theta, \phi)} \vec{u}(\theta, \phi) \left( \frac{\partial \vec{u}(\theta, \phi)}{\partial \theta^{(\mu)}} \right)^T \right) + Tr \left( \left( \frac{\partial \vec{u}(\theta, \phi)}{\partial \theta^{(\mu)}} \right) \cdot \vec{u}(\theta, \phi)^T \right)$$

$$- Tr \left( \frac{\vec{u}(\theta, \phi)^T \cdot \frac{\partial \vec{u}(\theta, \phi)}{\partial \theta^{(\mu)}}}{1 + \vec{u}(\theta, \phi)^T \cdot \vec{u}(\theta, \phi)} \vec{u}(\theta, \phi) \cdot \vec{u}(\theta, \phi)^T \right)$$

$$= \frac{\vec{u}(\theta, \phi)^T \cdot \left( \frac{\partial \vec{u}(\theta, \phi)}{\partial \theta^{(\mu)}} \right)}{\vec{u}(\theta, \phi)^T \cdot \vec{u}(\theta, \phi)} + \vec{u}(\theta, \phi)^T \cdot \left( \frac{\partial \vec{u}(\theta, \phi)}{\partial \theta^{(\mu)}} \right)$$

$$- \frac{\vec{u}(\theta, \phi)^T \cdot \vec{u}(\theta, \phi)}{1 + \vec{u}(\theta, \phi)^T \cdot \vec{u}(\theta, \phi)} \vec{u}(\theta, \phi)^T \cdot \frac{\partial \vec{u}(\theta, \phi)}{\partial \theta^{(\mu)}}$$

$$= \frac{2}{1 + \vec{u}(\theta, \phi)^T \cdot \vec{u}(\theta, \phi)} \vec{u}(\theta, \phi)^T \cdot \frac{\partial \vec{u}(\theta, \phi)}{\partial \theta^{(\mu)}}$$

Using the above equation, we have

$$\epsilon = \sqrt{g} \sum_{\mu=1}^{n} \left( \frac{\partial \vec{J}^{(\mu)}}{\partial \theta^{(\mu)}} + \frac{\vec{J}^{(\mu)}}{1 + \vec{u}(\theta, \phi)^T \cdot \vec{u}(\theta, \phi)} \sum_{\nu=1}^{n} \vec{u}^{(\nu)}(\theta) \frac{\partial \vec{u}^{(\nu)}(\theta)}{\partial \theta^{(\mu)}} \right)$$

This proves Proposition 1 below

$$Div(J^a)|_\theta = \frac{1}{\sqrt{g}} \epsilon$$

$$= \sum_{\mu=1}^{n} \left( \frac{\partial \vec{J}^{(\mu)}}{\partial \theta^{(\mu)}} + \frac{\vec{J}^{(\mu)}}{1 + \vec{u}(\theta, \phi)^T \cdot \vec{u}(\theta, \phi)} \sum_{\nu=1}^{n} \vec{u}^{(\nu)}(\theta) \frac{\partial \vec{u}^{(\nu)}(\theta)}{\partial \theta^{(\mu)}} \right)$$

## APPENDIX C

This appendix presents a proof of Proposition 2. For any $A \in \mathcal{SO}(n)$,

$$\exp(A) = I_n + A + \frac{1}{2!}A^2 + \frac{1}{3!}A^3 + \dots$$

Through SVD decomposition of A, we obtain

$$A = U \cdot \Sigma \cdot V^T$$

with $U$ and $V$ being two $n \times n$ unitary matrices. $\Sigma = Diag(\vec{\sigma})$ is a diagonal matrix. Therefore,

$$\exp(A) = I_n + U \cdot \Sigma \cdot V^T + \frac{1}{2!}(U \cdot \Sigma \cdot V^T)(U \cdot \Sigma \cdot V^T) + \frac{1}{3!}(U \cdot \Sigma \cdot V^T)(U \cdot \Sigma \cdot V^T)(U \cdot \Sigma \cdot V^T) + \dots$$

Note that $A^T = -A$, hence

$$(U \cdot \Sigma \cdot V^T)^T = -U \cdot \Sigma \cdot V^T = V \cdot \Sigma \cdot U^T$$

Consequently, $\forall k \geq 1$

$$A^k = \begin{cases} (-1)^{k/2} U \cdot \Sigma^k \cdot U^T, & k \text{ is even}; \\ (-1)^{(K+1)/2} V \cdot \Sigma^k \cdot U^T, & k \text{ is odd}. \end{cases}$$

In line with the above, we have

$$\exp(A) = \sum_{k=0}^{\infty} \frac{(-1)^k}{(2k)!} U \cdot \Sigma^{2k} \cdot U^T - \sum_{k=0}^{\infty} \frac{(-1)^k}{(2k+1)!} V \cdot \Sigma^{2k+1} \cdot U^T$$

$$= U \cdot \begin{bmatrix} cos(\vec{\sigma}^{(1)}) & 0 & 0 \\ 0 & \ddots & 0 \\ 0 & 0 & cos(\vec{\sigma}^{(n)}) \end{bmatrix} \cdot U^T - V \cdot \begin{bmatrix} sin(\vec{\sigma}^{(1)}) & 0 & 0 \\ 0 & \ddots & 0 \\ 0 & 0 & sin(\vec{\sigma}^{(n)}) \end{bmatrix} \cdot U^T$$

$$= U \cdot \Sigma_c \cdot U^T - V \cdot \Sigma_s \cdot U^T$$

This proves Proposition 2.

## APPENDIX D

This appendix presents a proof of Proposition 3. Following the assumption that $\exp(A) = \hat{\Omega} \cdot \Sigma_c \cdot \hat{\Omega}^T - \hat{\Phi} \cdot \Sigma_s \cdot \hat{\Omega}^T$, for any vector $\vec{a}$, we have

$$\exp(A) \cdot \vec{a} = \hat{\Omega} \cdot \Sigma_c \cdot \hat{\Omega}^T \cdot \vec{a} - \hat{\Phi} \cdot \Sigma_s \cdot \hat{\Omega}^T \cdot \vec{a}$$

Using Fourier transformation, we can re-write vector $\vec{a}$ in the Fourier series form below:

$$\vec{a}^{(j)} = \eta_0 + \sqrt{\frac{2}{n}} \sum_{i=1}^{n} \left[ \eta_i cos\left(\frac{2\pi i}{n}j\right) + \tilde{\eta}_i sin\left(\frac{2\pi i}{n}j\right) \right]$$

Hence,

$$\hat{\Omega}^T \cdot \vec{a} = \begin{bmatrix} \eta_1 \\ \vdots \\ \eta_n \end{bmatrix}$$

where $\eta_i = (\vec{\hat{\Omega}}^{(i)})^T \cdot \vec{a}$. Subsequently,

$$\Sigma_c \cdot \hat{\Omega}^T \cdot \vec{a} = \begin{bmatrix} cos(\vec{\sigma}^{(1)})\eta_1 \\ \vdots \\ cos(\vec{\sigma}^{(n)})\eta_n \end{bmatrix} \text{ and } \Sigma_s \cdot \hat{\Omega}^T \cdot \vec{a} = \begin{bmatrix} sin(\vec{\sigma}^{(1)})\eta_1 \\ \vdots \\ sin(\vec{\sigma}^{(n)})\eta_n \end{bmatrix}$$

Therefore,

$$\hat{\Omega} \cdot \Sigma_c \cdot \hat{\Omega}^T \cdot \vec{a} = [\vec{\hat{\Omega}}^{(1)}, \dots, \vec{\hat{\Omega}}^{(1)}] \cdot \begin{bmatrix} cos(\vec{\sigma}^{(1)})\eta_1 \\ \vdots \\ cos(\vec{\sigma}^{(n)})\eta_n \end{bmatrix}$$

$$= \sum_{j=1}^{n} cos(\vec{\sigma}^{(j)})\eta_j \vec{\hat{\Omega}}^{(j)}$$

$$\hat{\Phi} \cdot \Sigma_c \cdot \hat{\Omega}^T \cdot \vec{a} = [\vec{\hat{\Phi}}^{(1)}, \dots, \vec{\hat{\Phi}}^{(1)}] \cdot \begin{bmatrix} sin(\vec{\sigma}^{(1)})\eta_1 \\ \vdots \\ sin(\vec{\sigma}^{(n)})\eta_n \end{bmatrix}$$

$$= \sum_{j=1}^{n} sin(\vec{\sigma}^{(j)})\eta_j \vec{\hat{\Phi}}^{(j)}$$

We can now re-write $\exp(A) \cdot \vec{a}$ as

$$\exp(A) \cdot \vec{a} = \sum_{i=1}^{n} \eta_i \left( cos(\vec{\sigma}^{(i)})\vec{\hat{\Omega}}^{(i)} - sin(\vec{\sigma}^{(i)})\vec{\hat{\Phi}}^{(i)} \right)$$

$$= \sqrt{\frac{2}{n}} \sum_{i=1}^{n} \eta_i \begin{bmatrix} cos(\vec{\sigma}^{(i)})cos\left(\frac{2\pi i}{n}j\right)|_{j=0} - sin(\vec{\sigma}^{(i)})sin\left(\frac{2\pi i}{n}j\right)|_{j=n-1} \\ \vdots \\ cos(\vec{\sigma}^{(i)})cos\left(\frac{2\pi i}{n}j\right)|_{j=0} - sin(\vec{\sigma}^{(i)})sin\left(\frac{2\pi i}{n}j\right)|_{j=n-1} \end{bmatrix}$$

$$= \sqrt{\frac{2}{n}} \sum_{i=1}^{n} \eta_i \begin{bmatrix} cos\left(\frac{2\pi i}{n}j + \vec{\sigma}^{(i)}\right)|_{j=0} \\ \vdots \\ cos\left(\frac{2\pi i}{n}j + \vec{\sigma}^{(i)}\right)|_{j=n-1} \end{bmatrix}$$

In other words,

$$(\exp A \cdot \vec{a})^{(j)} = \sqrt{\frac{2}{n}} \sum_{i=1}^{n} \eta_i cos\left(\frac{2\pi i}{n}j + \vec{\sigma}^{(i)}\right)$$

This concludes that, upon multiplying $exp(A)$ with vector $\vec{a}$, it will lead to independent phase shifts of all frequency components of $\vec{a}$. In other words, rotating the $i$-th frequency component is equivalent to a phase shift of $\vec{\sigma}^{(i)}$ on that frequency component. This ends the proof of Proposition 3.

## APPENDIX E

This appendix presents a proof of Proposition 4. Any geodesic that passes through $\theta$ in manifold $(\mathbb{R}^n, g_{ab})$ and has $J^a|_\theta$ as its tangent vector at $\theta$ can be uniquely determined by the *geodesic equation* below Kreyszig (2013):

$$\frac{d^2\theta^{(\mu)}(t)}{dt^2} + \sum_{\nu=1}^{n}\sum_{\delta=1}^{n} \Gamma^\mu_{\nu\delta} \frac{d\theta^{(\nu)}(t)}{dt}\frac{d\theta^{(\delta)}(t)}{dt} = 0, \mu = 1, \dots, n$$

where $t$ stands for the geodesic parameter such that $\theta^{(\mu)}(0) = \theta^{(\mu)}$. $\Gamma^\mu_{\nu\delta}$ or $\Gamma^a_{bc}$ in the abstract index notation is the *Christoff symbol*. Therefore,

$$\frac{d^2\theta^{(\mu)}(t)}{dt^2} = -\sum_{\nu=1}^{n}\sum_{\delta=1}^{n} \Gamma^\mu_{\nu\delta} \frac{d\theta^{(\nu)}(t)}{dt}\frac{d\theta^{(\delta)}(t)}{dt}$$

subject to the conditions

$$\left(\frac{d\theta^{(\nu)}(t)}{dt}\right)|_{t=0} = \vec{J}^{(\nu)}, \nu = 1, \dots, n$$

Hence, updating $\theta$ along the direction of the geodesic can be approximated by the following learning rule:

$$\theta^{(\mu)} \leftarrow \theta^{(\mu)} + \alpha \left( \frac{\mathrm{d}\theta^{(\mu)}(t)}{\mathrm{d}t} \right)|_{t=0} - \alpha \Delta t \sum_{\nu=1}^{n} \sum_{\delta=1}^{n} \left[ \Gamma_{\nu\delta}^{\mu} \left( \frac{\mathrm{d}\theta^{(\nu)}(t)}{\mathrm{d}t} \right)|_{t=0} \left( \frac{\mathrm{d}\theta^{(\delta)}(t)}{\mathrm{d}t} \right)|_{t=0} \right]$$

where $\alpha$ is the learning rate. $\Delta t$ refers to a small increment of the geodesic parameter at $t = 0$. In view of the above, the geodesic regularized policy gradient can be approximated as

$$\vec{T}^{(\mu)} \approx \vec{J}^{(\mu)} - \Delta t \sum_{\nu=1}^{n} \sum_{\delta=1}^{n} \Gamma_{\nu\delta}^{\mu} \vec{J}^{(\nu)} \vec{J}^{(\delta)}$$

Because

$$\Gamma^{\delta}{}_{\mu\nu} = \Gamma_{ab}^{c} (\mathrm{d}\theta^{\delta})_c \left( \frac{\partial}{\partial \theta^{(\mu)}} \right)^a \left( \frac{\partial}{\partial \theta^{(\nu)}} \right)^b$$

$$= \frac{1}{2} \sum_{\rho=1}^{n} g^{\delta\rho} \left( \frac{\partial g_{\nu\rho}}{\partial \theta^{(\mu)}} + \frac{\partial g_{\mu\rho}}{\partial \theta^{(\nu)}} \right) - \frac{1}{2} \sum_{\rho=1}^{n} g^{\delta\rho} \left( \frac{\partial g_{\mu\nu}}{\partial \theta^{(\rho)}} \right)$$

We can study the two summations in the above equation separately. Let us denote

$$P1(\Gamma^{\delta}{}_{\mu\nu}) = \frac{1}{2} \sum_{\rho=1}^{n} g^{\delta\rho} \left( \frac{\partial g_{\nu\rho}}{\partial \theta^{(\mu)}} + \frac{\partial g_{\mu\rho}}{\partial \theta^{(\nu)}} \right)$$

$$P2(\Gamma^{\delta}{}_{\mu\nu}) = \frac{1}{2} \sum_{\rho=1}^{n} g^{\delta\rho} \left( \frac{\partial g_{\mu\nu}}{\partial \theta^{(\rho)}} \right)$$

Consequently,

$$\sum_{\mu=1}^{n} \sum_{\nu=1}^{n} P1(\Gamma^{\delta}{}_{\mu\nu}) \frac{\mathrm{d}\theta^{(\mu)}(t)}{\mathrm{d}t} \frac{\mathrm{d}\theta^{(\nu)}(t)}{\mathrm{d}t} = \frac{1}{2} \sum_{\mu=1}^{n} \sum_{\nu=1}^{n} \sum_{\rho=1}^{n} g^{\delta\rho} \left( \frac{\partial g_{\nu\rho}}{\partial \theta^{(\mu)}} + \frac{\partial g_{\mu\rho}}{\partial \theta^{(\nu)}} \right) \frac{\mathrm{d}\theta^{(\mu)}(t)}{\mathrm{d}t} \frac{\mathrm{d}\theta^{(\nu)}(t)}{\mathrm{d}t}$$

$$= \frac{1}{2} \sum_{\rho=1}^{n} g^{\delta\rho} \left( \sum_{\mu=1}^{n} \sum_{\nu=1}^{n} \left( \frac{\partial g_{\nu\rho}}{\partial \theta^{(\mu)}} + \frac{\partial g_{\mu\rho}}{\partial \theta^{(\nu)}} \right) \frac{\mathrm{d}\theta^{(\mu)}(t)}{\mathrm{d}t} \frac{\mathrm{d}\theta^{(\nu)}(t)}{\mathrm{d}t} \right)$$

Note that

$$\sum_{\mu=1}^{n} \sum_{\nu=1}^{n} \left( \frac{\partial g_{\nu\rho}}{\partial \theta^{(\mu)}} + \frac{\partial g_{\mu\rho}}{\partial \theta^{(\nu)}} \right) \frac{\mathrm{d}\theta^{(\mu)}(t)}{\mathrm{d}t} \frac{\mathrm{d}\theta^{(\nu)}(t)}{\mathrm{d}t} = 2 \sum_{\nu=1}^{n} \left( \sum_{\mu=1}^{n} \frac{\partial g_{\nu\rho}}{\partial \theta^{(\mu)}} \frac{\mathrm{d}\theta^{(\mu)}(t)}{\mathrm{d}t} \right) \frac{\mathrm{d}\theta^{(\nu)}(t)}{\mathrm{d}t}$$

In fact $\sum_{\mu=1}^{n} \frac{\partial g_{\nu\rho}}{\partial \theta^{(\mu)}} \frac{\mathrm{d}\theta^{\mu}}{\mathrm{d}t}$ captures the change of $g_{ab}$ along the direction of the geodesic. In view of this, because $g_{ab}$ **is assumed to change smoothly and stably along the geodesic**, we have

$$\Delta t \sum_{\mu=1}^{n} \frac{\partial g_{\nu\rho}}{\partial \theta^{(\mu)}} \frac{\mathrm{d}\theta^{(\mu)}(t)}{\mathrm{d}t} \approx -\zeta_1 \cdot g_{\nu\rho}, \zeta_1 > 0$$

Following the above,

$$\Delta t \sum_{\mu=1}^{n} \sum_{\nu=1}^{n} P1(\Gamma^{\delta}{}_{\mu\nu}) \left( \frac{\mathrm{d}\theta^{\mu}(t)}{\mathrm{d}t} \right)|_{t=0} \left( \frac{\mathrm{d}\theta^{\nu}(t)}{\mathrm{d}t} \right)|_{t=0} \approx \sum_{\rho=1}^{n} g^{\delta\rho} \sum_{\nu=1}^{n} (-\zeta_1 \cdot g_{\rho\nu}) \left( \frac{\mathrm{d}\theta^{\nu}(t)}{\mathrm{d}t} \right)|_{t=0}$$

$$= -\zeta_1 \left( \frac{\mathrm{d}\theta^{\delta}(t)}{\mathrm{d}t} \right)|_{t=0}$$

$$= -\zeta_1 \vec{J}^{(\delta)}$$

Accordingly,

$$\vec{T}^{(\sigma)} \approx \vec{J}^{(\sigma)} + \zeta_1 \vec{J}^{(\sigma)} + \frac{\Delta t}{2} \sum_{\rho=1}^{n} g^{\delta\rho} \sum_{\mu=1}^{n} \sum_{\nu=1}^{n} \left( \frac{\partial g_{\mu\nu}}{\partial \theta^{(\rho)}} \right) \left( \frac{\mathrm{d}\theta^{\mu}(t)}{\mathrm{d}t} \right)|_{t=0} \left( \frac{\mathrm{d}\theta^{\nu}(t)}{\mathrm{d}t} \right)|_{t=0}$$

Let $\zeta_2 = \frac{\Delta t}{2}$, we have

$$\vec{T}^{(\sigma)} \approx \vec{J}^{(\sigma)} + \zeta_1 \vec{J}^{(\sigma)} + \zeta_2 \sum_{\rho=1}^{n} g^{\delta\rho} \sum_{\mu=1}^{n} \sum_{\nu=1}^{n} \left( \frac{\partial g_{\mu\nu}}{\partial \theta^{(\rho)}} \right) \vec{J}^{(\mu)} \vec{J}^{(\nu)}$$

This proves Proposition 4. We can also re-write the above equation in the form of a matrix expression below for easy implementation by a deep learning library.

$$\vec{T} \approx (1 + \zeta_1)\vec{J} + \zeta_2 G_\theta^{-1} \cdot \nabla_\theta \left( NoGrad(\vec{J})^T \cdot G_\theta \cdot NoGrad(\vec{J}) \right)$$

Here, $NoGrad(\vec{J})$ indicates that vector $\vec{J}$ will not participate in the gradient calculation. $\nabla_\theta$ stands for the ordinary gradient operator w.r.t. $\theta$. Using the approximated $\vec{T}$, we can build a new learning rule below:

$$\theta \leftarrow \theta + \alpha \vec{T}$$

In line with this learning rule, $\frac{\zeta_2}{1+\zeta_1} > 0$ is treated as a hyper-parameter of the $g_{ab}$ regularization algorithm.

## APPENDIX F

In this appendix, we first introduce the new $g_{ab}$ regularization algorithm designed to train the metric tensor DNN model of $g_{ab}$ towards achieving close-to-zero divergence on $J^a$ at any $\theta$ of manifold $(\mathbb{R}^n, g_{ab})$. The pseudo-code of this algorithm is presented in Algorithm 1.

---

**Algorithm 1** The Metric Tensor Regularization Algorithm

---

Based on the up-to-date $\theta$, compute conventional policy gradient $\nabla_\theta \mathbb{E}_{s_0}[v_{\pi_\theta}(s_0)]$;
Using the metric tensor DNN, compute $J^a|_\theta$ and $Div(J^a)|_\theta$ using equation 8 and Proposition 1;
**while** the maximum number of iterations has not been reached **do**
    Update $\phi$ of the metric tensor DNN towards minimizing $(Div(J^a)|_\theta)^2$;
    Re-compute $J^a|_\theta$ and $Div(J^a)|_\theta$.
**end while**
Re-compute $J^a|_\theta$ based on the trained metric tensor DNN;
Compute geodesic regularized policy gradient $\vec{T}|_\theta$ using Proposition 4;
Return $J^a|_\theta$ and $\vec{T}|_\theta$.

---

**Algorithm 2** The Metric Tensor Regularized Policy Gradient Algorithm

---

Initialize policy network $\pi_\theta$ with randomly sampled $\theta \in \mathbb{R}^n$;
**for** each sampled episode till the maximum number of episodes is reached **do**
    Store all sampled state transitions into the replay buffer;
    Randomly sample a mini-batch from the replay buffer;
    Compute conventional policy gradient by using SAC, TD3 or other policy gradient algorithms;
    Compute $g_{ab}$ regularized and geodesic regularized policy gradients by using Algorithm 1;
    Train policy network $\pi_\theta$ by using regularized policy gradients.
**end for**
Return the trained policy network $\pi_\theta$.

---

Algorithm 1 starts from calculating the conventional policy gradient $\nabla_\theta \mathbb{E}_{s_0}[v_{\pi_\theta}(s_0)]$ w.r.t. the most recently learned policy parameter $\theta$. This can be achieved by using various existing DRL algorithms such as SAC and TD3. Afterwards, based on the metric tensor DNN, we compute the $g_{ab}$ regularized policy gradient vector $J^a$ as well as its divergence at $\theta$ by using equation 8 and Proposition 1 respectively. Guided by the square of the computed divergence as the loss function, Algorithm 1 updates the trainable parameters $\phi = \{\phi_1, \phi_2\}$ of the metric tensor DNN towards achieving close-to-zero divergence at $\theta$[3]. Using the trained metric tensor DNN, the $g_{ab}$ regularized policy gradient

---

[3]We set the maximum number of training iterations in Algorithm 1 to 20 in the experiments. We can further increase this number but it does not seem to produce any noticeable performance gains.

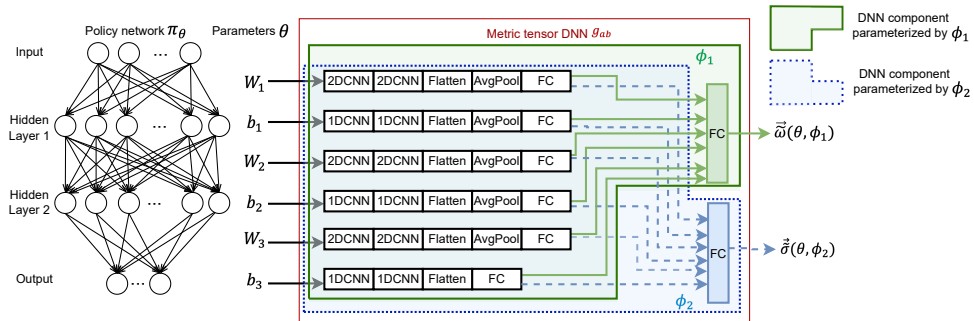

Figure 3: The metric tensor DNN designed to learn the metric tensor field $g_{ab}$ of the policy parametric space. The metric tensor DNN processes a policy network $\pi_\theta$ as its input and produces $\vec{\tilde{\omega}}(\theta, \phi_1)$ and $\vec{\tilde{\sigma}}(\theta, \phi_2)$ as its output. $\phi_1$ and $\phi_2$ together define the trainable parameters of the metric tensor DNN. Specifically, we denote the parameters contained within the green hexagon as $\phi_1$ and those within the blue hexagon as $\phi_2$. The intersection of these two hexagons corresponds to the common parameters shared between $\phi_1$ and $\phi_2$ in the metric tensor DNN. $W_i$ and $b_i$ refer respectively to the weight matrix and the bias vector of the $i$-th layer of the policy network $\pi_\theta$.

and the geodesic regularized policy gradient will be computed by Algorithm 1 as its output. These gradients will be utilized subsequently to train the policy network $\pi_\theta$.

Building on Algorithm 1, Algorithm 2 further presents a high-level description of $g_{ab}$ regularized DRL algorithms. Algorithm 2 is designed to be compatible with SAC, TD3 and many other policy gradient algorithms. In fact, it is straightforward to extend SAC and TD3 to build their $g_{ab}$ regularized counterparts. Without modifying any existing operations of SAC and TD3, at every iteration of training $\pi_\theta$, we can perform Algorithm 1 to train the metric tensor DNN and compute the corresponding metric tensor regularized policy gradients. Subject to the algorithm variants, either the $g_{ab}$ regularized policy gradient or the geodesic regularized policy gradient will be computed and utilized to train the policy network $\pi_\theta$. The trained $\pi_\theta$ is then exploited to collect new state-transition samples. The above process is repeatedly executed until a certain number of state-transition samples have been collected to train $\pi_\theta$.

## APPENDIX G

To learn the complex geometric structure of $g_{ab}$, we introduce a new architecture for the metric tensor DNN. This is exemplified by an example *metric tensor DNN* for a policy network $\pi_\theta$ with two hidden layers, as depicted in Figure 3. The metric tensor DNN parameterized by $\phi_1$ and $\phi_2$ maps the $n$-dimensional policy parameter $\theta = [W_1, b_1, ..., W_3, b_3]$ into two $\tilde{m}$-dimensional vectors $\vec{\tilde{\omega}}(\theta, \phi_1)$ and $\vec{\tilde{\sigma}}(\theta, \phi_2)$, which are used to build the scaling matrix $S(\theta, \phi_1)$ and the rotation matrix $R(\theta, \phi_2)$ in equation 3 respectively[4].

In particular, each layer of weight matrix $W_i$ and bias vector $b_i$ of $\pi_\theta$ is processed individually through two consecutive convolutional kernels (2D kernels of size 3×3 for processing $W_i$ or 1D kernels of size 3 for processing $b_i$), followed by the flattening and average pooling operations with a pool size of 5 before passing through the first dense layer in Figure 3. It should be noted that the bias vector for the output layer is exempt from the pooling operation due to its comparatively low dimensionality[5]. The *Softplus* function serves as the activation mechanism for both the convolutional and dense layers in the metric tensor DNN. We also used the ReLU activation function and obtained

---

[4]On LunnarLanderContinuous-v2, the input dimension $n$ of the metric tensor DNN is 69124 for a policy with two hidden layers described in Table 2 for SAC-T. The output dimension is 700, effectively yielding two $\tilde{m}$-dimensional vectors ($\tilde{m}$=350).

[5]Note that the dimensionality of the bias vector for the output layer of the policy network $\pi_\theta$ equals to the dimensionality of the action space, which is usually small. For example, the dimensionality of the action space is 2 for the LunnarLanderContinuous-v2 problem.

similar experiment results. The outputs of the first dense layer are concatenated and channeled through two separate and additional dense layers in Figure 3, each of which yields an $\tilde{m}$-dimensional vector.

While the performance of the metric tensor DNN could be further enhanced by fine-tuning the DNN architecture in Figure 3, such an undertaking is beyond the scope of this paper. Consequently, we reserve the exploration of more advanced network architecture designs and fine-tuning for our future work. Moreover, the performance of the learned metric tensor DNN reported in Section 6 shows that our proposed simple architecture for the metric tensor DNN can effectively learn $g_{ab}|_\theta$ w.r.t. any policy parameter $\theta$ such that the absolute divergence at $\theta$ can be reduced.

## APPENDIX H

This appendix presents the detailed experiment setup. We use the popular OpenAI Spinning Up repository Achiam (2018) to implement $g_{ab}$ regularized DRL algorithms proposed in this paper. Our implementation follows closely all hyper-parameter setting and network architectures reported in Haarnoja et al. (2018); Fujimoto et al. (2018) and summarized in Table 2. Since calculating the Hessian trace precisely can pose significant computation burden on existing deep learning libraries such as PyTorch, we adopt a popular Python library named PyHessian Yao et al. (2020), where Hutchinson's method Avron & Toledo (2011); Bai et al. (1996) is employed to estimate the Hessian trace efficiently. All experiments were conducted on a cluster of Linux computing nodes with 2.5 GHz Intel Core i7 11700 processors and 16 GB memory. To ensure consistency, all experiments were run in a virtual environment with Python 3.7.11 managed by the Anaconda platform. The main Python packages used in our experiments are summarized in Table 3.

Table 2: Hyper-parameter settings of all experimented algorithms.

| Hyper-parameter | SAC | SAC-J | SAC-T | TD3 | TD3-J | TD3-T |
|---|---|---|---|---|---|---|
| Total training timesteps | 300,000 | 300,000 | 300,000 | 300,000 | 300,000 | 300,000 |
| Max episode length | 1000 | 1000 | 1000 | 1000 | 1000 | 1000 |
| Minibatch size | 256 | 256 | 256 | 100 | 100 | 100 |
| Adam learning rate | 3e-4 | 3e-4 | 3e-4 | 1e-3 | 1e-3 | 1e-3 |
| Discount ($\gamma$) | 0.99 | 0.99 | 0.99 | 0.99 | 0.99 | 0.99 |
| GAE parameter ($\lambda$) | 0.995 | 0.995 | 0.995 | 0.995 | 0.995 | 0.995 |
| Replay buffer size | 1e6 | 1e6 | 1e6 | 1e6 | 1e6 | 1e6 |
| Update interval (timesteps) | 50 | 50 | 50 | 50 | 50 | 50 |
| Network architecture | 256x256 | 256x256 | 256x256 | 400x300 | 400x300 | 400x300 |

Table 3: Python packages.

| Package name | Version |
|---|---|
| cython | 0.29.25 |
| gym | 0.21.0 |
| mujoco-py | 2.1.2.14 |
| numpy | 1.21.4 |
| pybulletgym | 0.1 |
| python | 3.7.11 |
| PyHessian | 0.1 |
| torch | 1.13.1 |

## APPENDIX I

Figure 4 and Figure 5 present the divergence ratios obtained by SAC-J and TD3-J during the training process on four benchmark problems. As evidenced by the figures, using the trained metric tensor DNN, SAC-J and TD3-J can successfully reduce a significant portion of the divergence ratios to below 1 during the training process on all benchmark problems. Meanwhile, Table 4 confirms that over 70% of the divergence ratios obtained by SAC-J and TD3-J during policy training are less than 1 on all benchmark problems. The divergence ratios can be further reduced to below 50% with a good probability on majority of the experimented benchmark problems. These results demonstrate

the effectiveness of our metric tensor regularization algorithm in training the proposed metric tensor DNN towards achieving close-to-zero divergence on the manifold $(\mathbb{R}^n, g_{ab})$.

Table 4: The percentage of divergence ratios $< 1$ and $< 50\%$ for SAC-J and TD3-J on four benchmark problems.

| Benchmark problems | Divergence ratio < 1 (%) | | Divergence ratio < 0.5 (%) | |
|---|---|---|---|---|
| | SAC-J | TD3-J | SAC-J | TD3-J |
| InvertedDoublePendulum-v2 (Mujoco) | 89 | 87 | 50 | 53 |
| Walker2D-v3 (Mujoco) | 76 | 79 | 0 | 23 |
| Ant-v0 (PyBullet) | 70 | 71 | 19 | 29 |
| Walker2D-v0 (PyBullet) | 97 | 93 | 64 | 72 |

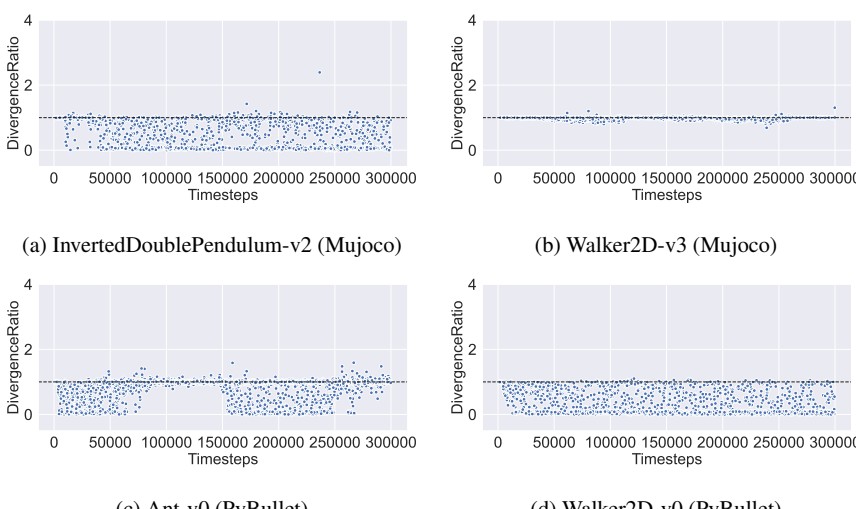

(a) InvertedDoublePendulum-v2 (Mujoco)

(b) Walker2D-v3 (Mujoco)

(c) Ant-v0 (PyBullet)

(d) Walker2D-v0 (PyBullet)

Figure 4: Divergence ratios obtained by SAC-J during the training process, where the divergence ratio is defined as the absolute ratio between $Div(J^a)$ and the Hessian trace.

## APPENDIX J

In this appendix, we present the Hessian trace observed during the policy training process for SAC and TD3 respectively, as shown in Figure 6. The Hessian trace results clearly show that the Hessian trace can differ substantially across different algorithms on the same benchmark problem. For example, the Hessian trace obtained by SAC-T and SAC-J on InvertedDoublePendulum-v2 is mostly in the value range between -1000 and 0. On the same benchmark, the Hessian trace obtained by TD3-T and TD3-J is in a significantly different value range between -3 and 2.

Similarly, Figure 6 demonstrates that the Hessiance trace can vary hugely for the same algorithm on different benchmarks. For instance, considering SAC-T and SAC-J, the Hessian trace obtained by the two SAC variants is normally in the value range between -150 and 0 on Ant-v0. However, the Hessian trace obtained by the same algorithms is in a substantially differed value range between -2000 and -500 on the Walker2D-v3 benchmark.

Based on the above observation, we believe that the influence of the Hessian trace on the performance of policy gradient algorithms should not be overlooked. It motivates us to develop $g_{ab}$ regularzied policy gradient algorithms in this paper.

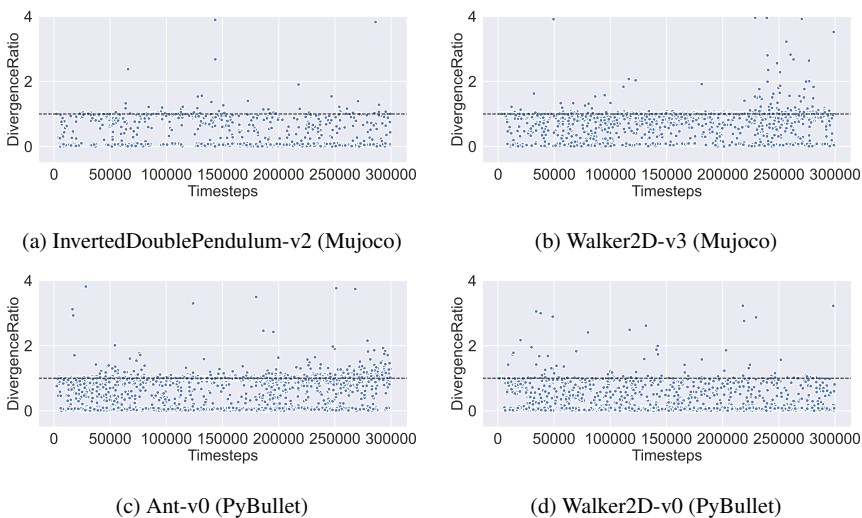

Figure 5: Divergence ratios obtained by TD3-J during the training process, where the divergence ratio is defined as the absolute ratio between $Div(J^a)$ and the Hessian trace.

## APPENDIX K

In this appendix, we investigate the performance impact of three hyper-parameters. They are (1) $\frac{\zeta_2}{1+\zeta_1}$ for estimating the geodesic regularized policy gradient $\vec{T}|_\theta$ in Proposition 4; (2) the number of iterations for the Hutchinson's method to approximate the Hessian trace Avron & Toledo (2011); Bai et al. (1996); and (3) the number of iterations for training the metric tensor DNN in Algorithm 1. SAC-T and Walker2D-v3 will be utilized in this appendix to demonstrate the performance impact of varied settings of these hyper-parameters. Similar observations have also been witnessed on other algorithm variants and benchmark problems, and will not be reported further.

**The estimation of $\vec{T}|_\theta$ in Proposition 4:** To investigate the performance impact of using the hyper-parameter $\frac{\zeta_2}{1+\zeta_1}$ to estimate $\vec{T}|_\theta$ in Proposition 4, we tested 2 different settings of $\frac{\zeta_2}{1+\zeta_1}$, including 0.1 and 0.3. The respective learning curves are plotted jointly in Figure 7(a). As shown in the figure, different settings of $\frac{\zeta_2}{1+\zeta_1}$ can achieve similar cumulative returns where $\frac{\zeta_2}{1+\zeta_1} = 0.1$ obtained slightly higher returns after 300k timesteps, in comparison to the case that $\frac{\zeta_2}{1+\zeta_1} = 0.3$. Therefore, we set $\frac{\zeta_2}{1+\zeta_1} = 0.1$ for the main experiment results reported in Section 6 of this paper.

**The number of iterations for the Hutchinson's method:** As explained in Appendix H, our implementation of Algorithm 1 adopted a Python library named PyHessian Yao et al. (2020) to efficiently estimate the Hessian trace associated with the policy gradient in the Euclidean policy parametric space. In particular, we used the Hutchinson's method to estimate the Hessian trace in an iterative manner Avron & Toledo (2011); Bai et al. (1996) where the number of iterations is a hyper-parameter that controls the estimation accuracy.

We evaluated two different numbers of iterations, including 30 and 50. The corresponding performance results are reported in Figure 7(b). We notice that the performance impact of different iteration numbers appears to be small. Intuitively, a larger number of iterations can estimate the Hessian trace more accurately, which may help to increase the cumulative returns and reduce the performance variance. However, it comes at a cost of higher computation burden. Since 30 iterations can achieve competitive final performance in our experiments, we recommend to set the iteration number for the Hutchinson's method to 30 in this paper to reduce the computation cost.

**The number of iterations for training the metric tensor DNN:** In this paper, a new metric tensor DNN is designed to learn the complex geometric structure of $g_{ab}$. As summarized in Algorithm 1, we

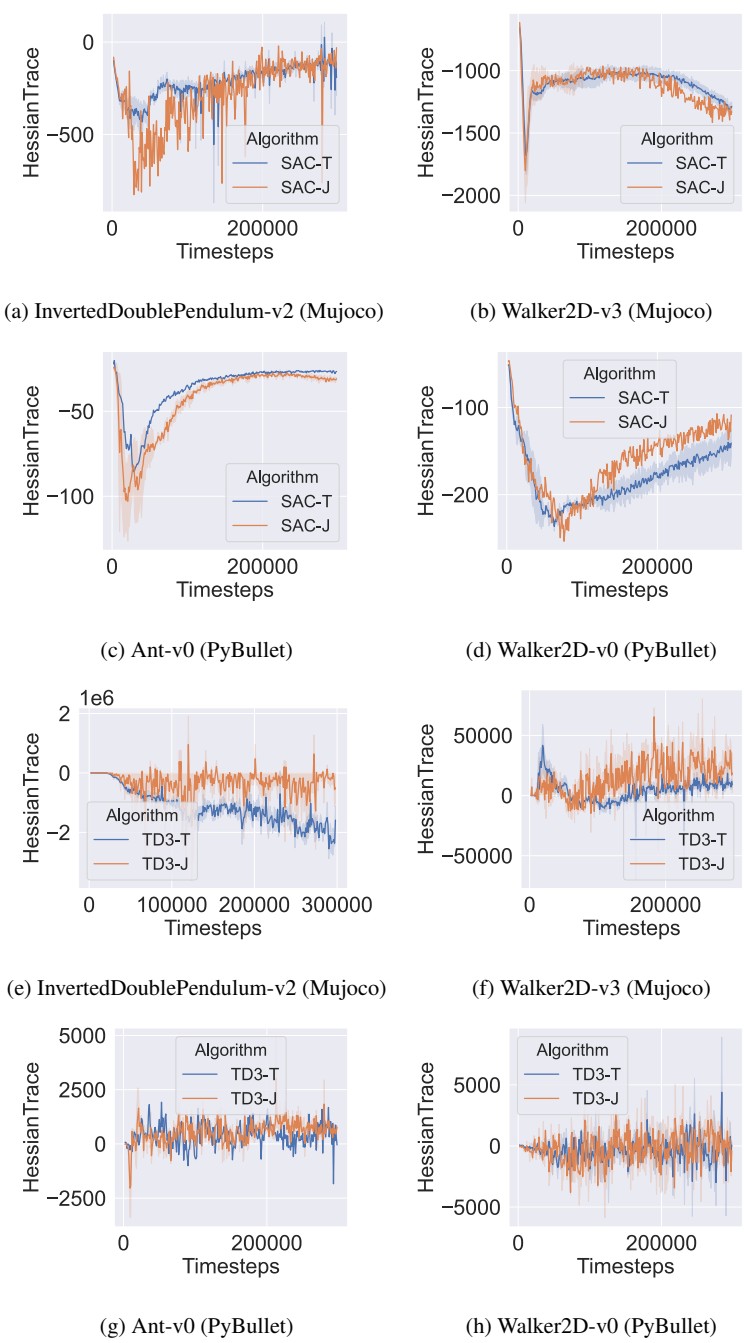

Figure 6: Hessian trace trend during the training process for (a)-(d) SAC-J and SAC-T and (e)-(f) TD3-J and TD3-T.

train the metric tensor DNN for a certain number of iterations with the goal of reducing the absolute divergence of the policy gradient vector field $J^a$ towards zero at any $\theta$ of the manifold $(\mathbb{R}^n, g_{ab})$.

Similar to the number of iterations for the Hutchinson's method, the number of iterations for training the metric tensor DNN also plays an important role of balancing the trade-off between the accuracy of

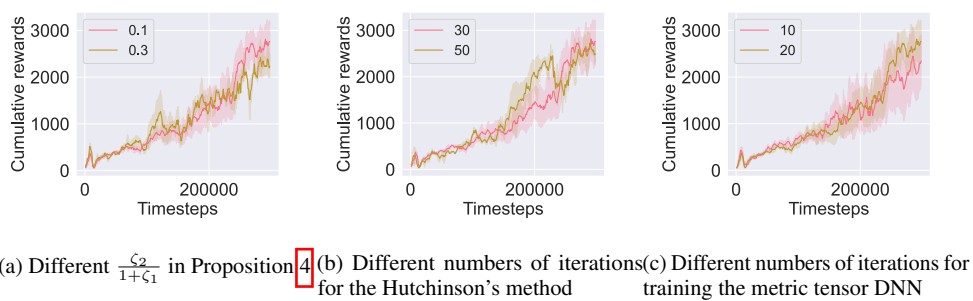

(a) Different $\frac{\zeta_2}{1+\zeta_1}$ in Proposition 4 (b) Different numbers of iterations for the Hutchinson's method (c) Different numbers of iterations for training the metric tensor DNN

Figure 7: The impact of using different hyper-parameters on the performance of SAC-T.

estimating $g_{ab}$ and the computation cost. In this appendix, we tested two different iteration numbers, as presented in Figure 7(c). It can be easily verified that, after 300k timesteps, more iterations can noticeably achieve higher cumulative returns and lower performance variance (brown shaded area). Thus, in the main experiment results reported in Section 6, the iteration number for training the metric tensor DNN is set to 20. Meanwhile, we notice that additional performance gains can be realized by further increasing this iteration number. However, in view of the extra computation cost, we believe 20 is a good setting for this hyper-parameter.