# OpenReview forum: "Deep Metric Tensor Regularized Policy Gradient"
_ICLR.cc/2024/Conference — ICLR 2024 Conference Withdrawn Submission_

### Official Review · Reviewer_dQEm · 2023-11-01

**Soundness:** 2 fair
**Presentation:** 1 poor
**Contribution:** 2 fair
**Rating:** 3
**Confidence:** 4

**Summary:**

This paper proposes a method to learn a Riemannian metric g_{ab} for use in minimizing divergences for policy gradient methods.

**Strengths:**

* The paper tackles an interesting question (how does the metric of the underlying policy parameter space affect the learning of policy gradients).
* The empirical results are somewhat promising, as the method seems to improve the policy gradient construction.

**Weaknesses:**

* The paper is not formatted correctly. In particular, the length of the paper sheet seem off (at least compared with my other submissions which can be checked by doing a fitted zoom), and the header of each page is missing the "submitted to ICLR" tag and seems too large. I'm not sure if this violates the submission guidelines as this does seem to be a noticeable but perhaps not extremely large deviation with the existing style guidelines.
* There are major issues with the current presentation. In particular, it is never explicitly made clear why $g$ should be learned in the introduction or in Section 3 or 4. There needs to be a bit more motivation for why this quantity can e.g. help optimization when it is not the standard Euclidean metric. On this same line, the connection between learning and the divergence operator should also be further explained, especially because this paper intersects two rather disparate communities.
* Continued: the construction is extremely convoluted and seems quite suboptimal. In particular, $g$ is defined as a low rank structure in equation 2, but then the low rank component is parameterized in an extremely strange manner (with rotation and scaling matrices). These other matrices must be further approximated with stuff like fourier coefficients, adding more computational/logistical overhead. Why not just parameterize $\vec{u}$ as an output of a neural network if this is what we need to learn? Also, this construction seems to be very weak, as the representation capability of rank-1 matrix fields is insufficient. Why not just parameterize the whole matrix (or some lower rank component) with the exponential map to SPD matrices (as is done in [1])?
* There are fundamental limitations with learning the metric tensor. In particular, this is somewhat high dimensional since it must be $O(n^2)$. I assume the paper deals with this with the low rank approximation, but further information on the experimental settings used in this paper could help shed light into whether or not the low rank approximation is required.

[1] https://openreview.net/forum?id=25HMCfbzOC

**Questions:**

N/A

---

### Official Review · Reviewer_WRcW · 2023-11-01

**Soundness:** 3 good
**Presentation:** 3 good
**Contribution:** 3 good
**Rating:** 6
**Confidence:** 1

**Summary:**

The paper proposes a novel policy gradient algorithm for deep reinforcement learning. It uses Hessian trace information within the parameter space of policy to boost the effectiveness of trained policy networks.

**Strengths:**

The paper Introduces a metric tensor field that maps the parameter space of policies into a broader Riemannian manifold
The paper conducts extensive experiments to prove the effectiveness of the proposed method.

**Weaknesses:**

The paper evaluates the effectiveness of the proposed method based on 4 benchmark problems, which is not comprehensive. The paper should conduct experiments on more datasets to have a more solid conclusion.

**Questions:**

Please refer to the weakness part.

---

### Official Review · Reviewer_72W5 · 2023-11-05

**Soundness:** 2 fair
**Presentation:** 1 poor
**Contribution:** 1 poor
**Rating:** 3
**Confidence:** 4

**Summary:**

This paper studies policy gradient and proposes a metric tensor regularization method to control the Hessian information during training.

**Strengths:**

1. The authors provide a novel perspective for regularization in policy gradient methods.
2. The paper does a good job summarizing the related works in policy gradient algorithms.

**Weaknesses:**

1. The paper is extremely hard to follow. The terminology and theoretical results largely differ from the ones in RL theory papers. Let me specify.
2. The authors did not provide much intuition before diving into the mathematical formulations. It is not clear why policy gradient methods should control the Hessian information. The authors claimed that "the Hessian information can vary substantially during the training of the
policy network". But I fail to see the evidence. And this needs more justification.
3. The usefulness of the provided theoretical results is also not clear. They are more like the general mathematical results under the manifold framework. The standard results in RL theory, such as sample efficiency, are missing from the analysis. The results themselves are also making me very confused. For example, Proposition 2 studies the Special Orthogonal (SO) group. What does it refer to in the policy gradient method? I fail to see if the results indicate higher efficiency or any other advantage of the proposed mechanism in policy gradient algorithms. I strongly encourage the authors to move the unnecessary theory results to the appendix and add more results for the proposed method.
4. It is not clear if the analysis in Riemannian manifold is valid, given that the policy space is relatively low for RL tasks considered (it might be high, but still much lower compared to high-dimensional analysis). This concern also remains regarding the experimental results.
5. The experimental results do not support the claims well. The results only compare to SAC and TD3, while the improvement is marginal. Besides, the baselines are lower than the original reported results in some environments. For example, the returns of SAC and TD3 reported in many other papers are above 5000 in the Mujoco walker2d task.
6. The regularization in parameter space is also not novel. The authors need to give more evidence of why policy gradient needs the proposed mechanism, especially given its complexity and marginal experimental improvement.

**Questions:**

See weakness above.